# Linear Combination of Saved Checkpoints Makes Consistency and Diffusion Models Better

**Enshu Liu**[*1,2], **Junyi Zhu**[*3], **Zinan Lin**[†‡4], **Xuefei Ning**[†‡1], **Shuaiqi Wang**[5],
**Matthew B. Blaschko**[3], **Sergey Yekhanin**[4], **Shengen Yan**[2], **Guohao Dai**[2,6],
**Huazhong Yang**[1], **Yu Wang**[‡1]

[1]Tsinghua University  [2]Infinigence-AI  [3]KU Leuven  [4]Microsoft Research
[5]Carnegie Mellon University  [6]Shanghai Jiao Tong University

## ABSTRACT

Diffusion Models (DM) and Consistency Models (CM) are two types of popular generative models with good generation quality on various tasks. When training DM and CM, intermediate weight checkpoints are not fully utilized and only the last converged checkpoint is used. In this work, we find proper checkpoint merging can significantly improve the training convergence and final performance. Specifically, we propose LCSC, a simple, effective, and efficient method to enhance the performance of DM and CM, by combining checkpoints along the training trajectory with coefficients deduced from evolutionary search. We demonstrate the value of LCSC through two use cases: **(a) Reducing training cost.** With LCSC, we only need to train DM/CM with fewer number of iterations and/or lower batch sizes to obtain comparable sample quality with the fully trained model. For example, LCSC achieves considerable training speedups for CM ($23\times$ on CIFAR-10 and $15\times$ on ImageNet-64). **(b) Enhancing pre-trained models.** When full training is already done, LCSC can further improve the generation quality or efficiency of the final converged models. For example, LCSC achieves better FID using 1 number of function evaluation (NFE) than the base model with 2 NFE on consistency distillation, and decreases the NFE of DM from 15 to 9 while maintaining the generation quality. Applying LCSC to large text-to-image models, we also observe clearly enhanced generation quality.

## 1 INTRODUCTION

Diffusion Models (DMs) (Sohl-Dickstein et al., 2015; Ho et al., 2020; Song et al., 2020b) as a generative modeling paradigm have rapidly gained widespread attention in the last several years, showing excellent performance in various tasks like image generation (Ho et al., 2020; Dhariwal & Nichol, 2021; Rombach et al., 2022), video generation (Ho et al., 2022; Blattmann et al., 2023), and 3D generation (Poole et al., 2022; Lin et al., 2023). DM requires an iterative denoising process in the generation process, which could be slow. Consistency Models (CMs) (Song et al., 2023) are proposed to handle this dilemma, which provides a better generation quality under one or few-step generation scenarios and is also broadly applied (Luo et al., 2023a;b; Wang et al., 2023).

In this paper, we investigate the training process of DM and CM. We find that checkpoints—the model weights periodically saved during the training process—have under-exploited potential in boosting the performance of DM and CM. In particular, within the metric landscape, we observe numerous high-quality basins located near any given point along the optimization trajectory. These high-quality basins *cannot* be reliably reached through Stochastic Gradient Descent (SGD), including its advanced variants like Adam (Kingma & Ba, 2014). However, we find that an appropriate linear combination of different checkpoints can locate these basins.

---

\* Co-first authors: Enshu Liu (`les23@mails.tsinghua.edu.cn`), Junyi Zhu (`junyizhu.ai@gmail.com`)

† Co-advise

‡ Corresponding authors: Yu Wang (`yu-wang@mail.tsinghua.edu.cn`), Xuefei Ning (`foxdoraame@gmail.com`), and Zinan Lin (`zinanlin@microsoft.com`).

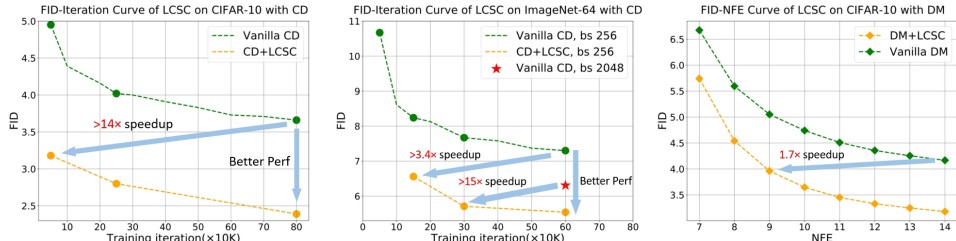

Figure 1: Comparison of LCSC and vanilla training. LCSC achieves more than 14× training speed up on CIFAR-10 with Consistency Distillation (CD) and more than 15× training speed up compared to official released model on ImageNet-64 with CD. LCSC can also enhance the final converged model significantly and achieves 1.7× inference speedup for DM.

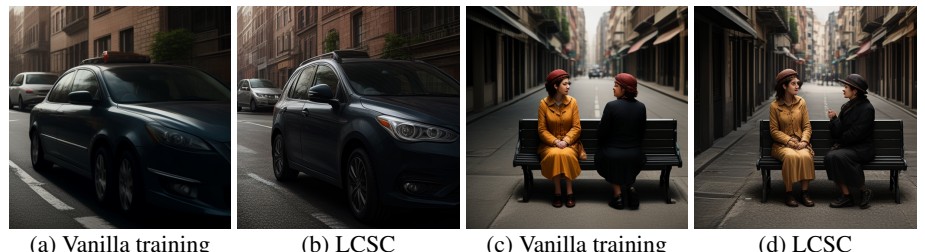

| (a) Vanilla training | (b) LCSC | (c) Vanilla training | (d) LCSC |

Figure 2: Comparison between the images generated by LCSC and vanilla training (on LCM-LoRA model (Luo et al., 2023a;b)). The prompts for the left and right images are "A car that seems to be parked illegally behind a legally parked car" and "Two women waiting at a bench next to a street" respectively. The vanilla method produces images with unrealistic front wheels (left) and an unnatural sitting posture (right), whereas LCSC produces more realistic images.

Inspired by the observation, we propose Linear Combination of Saved Checkpoints (LCSC). In particular, for a set of checkpoints saved until any point of the training process, LCSC searches for the optimal linear combination coefficients that optimize certain metrics (e.g., FID (Heusel et al., 2017)) using an evolutionary algorithm. LCSC optimizes only in a low-dimension space and needs no back-propagation, thus could be faster than training. Additionally, LCSC can optimize objectives whose gradients are hard to compute, under which gradient-based optimization is not applicable. We note that the widely used Exponential Moving Average (EMA) (Szegedy et al., 2016) in DM and CM can be viewed as a linear combination method whose combination coefficients are determined heuristically. We will theoretically prove and empirically demonstrate that EMA coefficients are sub-optimal and are worse than LCSC. While this paper focuses on DM and CM, from a broader perspective, LCSC is a general method and may be expanded to other tasks and neural networks.

**We argue that LCSC is beneficial in all steps in the DM/CM production stage and can be used to**: (a) Reduce training cost. The training process of DM and CM is very costly. On ImageNet-64, SOTA DMs and CMs (Dhariwal & Nichol, 2021; Karras et al., 2022; Song et al., 2023) require more than 10K GPU hours on Nvidia A100. For higher resolution tasks, the resource demand soars dramatically again: Stable Diffusion (Rombach et al., 2022) costs more than 150K GPU hours (approx. 17 years) on Nvidia A100. *By applying LCSC at the end, we can train CM/DM with significantly fewer iterations or smaller batch sizes and reach similar generation quality with the fully trained model, thereby reducing the computational cost of training.* (b) Enhance model performance. If the full training is already done, LCSC can still be applied to get a model that is *better than any model in the training process*. For model developers with saved checkpoints, LCSC can be directly applied. For users who can only access the final released checkpoint, they can fine-tune the checkpoint for a few more iterations and apply LCSC on these checkpoints. As DM/CM provides a flexible trade-off between generation quality and the number of generation steps, *the enhanced model from LCSC could lead to either better generation quality or faster generation.*

**The reminder of this paper is organized as follows:** In Sec. 2, we introduce the background and related works. In Sec. 3, we visualize the metric landscape of DM and CM, demonstrating the potential of checkpoint merging. Additionally, we provide theoretical analyses showing that EMA is suboptimal, and that a more flexible method for setting the merging coefficients is preferable. In Sec. 4, we present our efficient method LCSC, which flexibly adjusts the merging coefficients to find the best model through evolutionary search. A schematic diagram of LCSC is shown in Fig. 3 In Sec. 5, we conduct experiments to validate the effectiveness of LCSC on both DM and CM. The results show that LCSC can accelerate the training process of CM by up to 23×. Moreover, LCSC

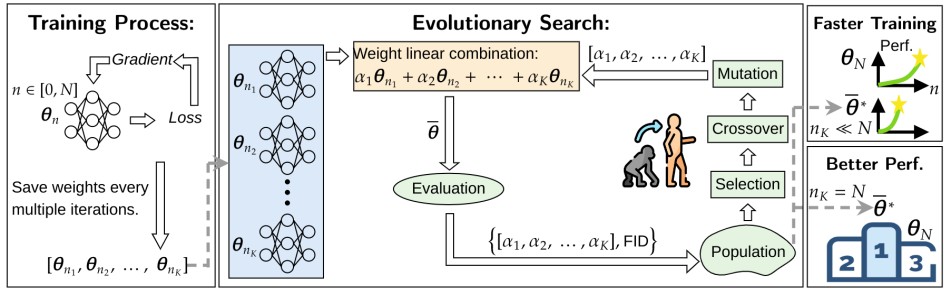

Figure 3: A schematic diagram of LCSC. Given a set of checkpoints from training (left), LCSC use evolutionary search to find the optimal linear combination coefficients (middle). LCSC can be applied on checkpoints from a training process with fewer training iterations or batch sizes and still gets similar performance (abbreviated as "Perf."), thus reducing training cost and enabling faster training. LCSC can also enhance the final model in terms of generation quality or speed (right).

can decrease the NFE of DM from 15 to 9 while keeping the sample quality. We highlight some of the results in Fig. 1. Additionally, we demonstrate that LCSC can be applied to text-to-image models to further improve generation quality, with some images presented in Fig. 2. In Sec. 6, we discuss the searched patterns of coefficients to inspire further research on checkpoints merging.

## 2 BACKGROUND AND RELATED WORK

In this section, we briefly introduce the foundational concepts and related works necessary for understanding this paper. A more comprehensive discussion is provided in App. B

**Diffusion Probabilistic Model.** Let us denote the data distribution by $p_{data}$, diffusion models (Sohl-Dickstein et al., 2015; Song et al., 2020a; Ho et al., 2020; Nichol & Dhariwal, 2021; Song et al., 2020b) learn a process that perturbs $p_{data}$ with a stochastic differential equation:

$$d\mathbf{x}_t = \boldsymbol{\mu}(\mathbf{x}_t, t)dt + \sigma(t)d\boldsymbol{w}_t, \tag{1}$$

where $\boldsymbol{\mu}(\cdot, \cdot)$ and $\sigma(\cdot)$ represent the drift and diffusion coefficients, respectively, $\boldsymbol{w}_t$ denotes the standard Brownian motion, and $t \in [0, T]$ indicates the time step. $t = 0$ stands for the real data distribution. $\boldsymbol{\mu}(\cdot, \cdot)$ and $\sigma(\cdot)$ are designed to make sure $p_T(\boldsymbol{x})$ becomes pure Gaussian noise.

To train a diffusion model, we can construct a network $\boldsymbol{s}_{\boldsymbol{\theta}}(\boldsymbol{x}_t, t)$ to approximate the score function of the perturbed data distribution $\nabla \log p_t(\boldsymbol{x})$ by minimizing:

$$\mathbb{E}_{t \sim \mathcal{U}(0,T]}\mathbb{E}_{\mathbf{y} \sim p_{\text{data}}}\mathbb{E}_{\mathbf{x}_t \sim \mathcal{N}(\mathbf{y}, \sigma(t)^2 \mathbf{I})} \lambda(t) \|\mathbf{s}_{\boldsymbol{\theta}}(\mathbf{x}_t, t) - \nabla_{\mathbf{x}_t} \log p(\mathbf{x}_t | \mathbf{y})\|, \tag{2}$$

where $\lambda(t)$ represents the loss weighting and $\boldsymbol{y}$ denotes a training image.

**Consistency Models.** During generation, diffusion models require multiple time steps, which results in low efficiency. Consistency models (CM) are proposed to enable single-step generation (Song et al., 2023), by mapping any point $\boldsymbol{x}_t$ at any given time $t$ along a probability flow trajectory directly to the trajectory's initial point $\boldsymbol{x}_t$. Training of CM can follow one of two methodologies: consistency distillation (CD) or direct consistency training (CT). In the case of CD, the model $f_{\boldsymbol{\theta}}$ leverages knowledge distilled from a pre-trained DM $\phi$. The distillation loss can be formulated as follows:

$$\mathbb{E}_{k \sim \mathcal{U}[1,K-1]}\mathbb{E}_{\boldsymbol{y} \sim p_{data}}\mathbb{E}_{\boldsymbol{x}_{t_{k+1}} \sim \mathcal{N}(\boldsymbol{y}, t_{k+1}^2 \boldsymbol{I})} \lambda(t_k)d(f_{\boldsymbol{\theta}}(\boldsymbol{x}_{t_{k+1}}, t_{k+1}), f_{\boldsymbol{\theta}^-}(\hat{\boldsymbol{x}}_{t_k}^{\phi}, t_k)), \tag{3}$$

where $f_{\boldsymbol{\theta}^-}$ refers to the target model and $\boldsymbol{\theta}^-$ is computed through EMA of the historical weights of $\boldsymbol{\theta}$, $\hat{\boldsymbol{x}}_{t_k}^{\phi}$ is estimated by the pre-trained diffusion model $\phi$ through one-step denoising based on $\boldsymbol{x}_{t_{k+1}}$, and $d$ is a metric such as $\ell_2$ distance. Alternatively, in the CT case, the models $f_{\boldsymbol{\theta}}$ are developed independently, without relying on any pre-trained DM:

$$\mathbb{E}_{k \sim \mathcal{U}[1,K-1]}\mathbb{E}_{\boldsymbol{y} \sim p_{data}}\mathbb{E}_{\boldsymbol{z} \sim \mathcal{N}(\boldsymbol{0}, \boldsymbol{I})} \lambda(t_k)d(f_{\boldsymbol{\theta}}(\boldsymbol{y} + t_{k+1}\boldsymbol{z}, t_{k+1}), f_{\boldsymbol{\theta}^-}(\boldsymbol{y} + t_k\boldsymbol{z}, t_k)), \tag{4}$$

where the target model $f_{\boldsymbol{\theta}^-}$ is set to be the same as the model $f_{\boldsymbol{\theta}}$ in the latest improved version of the consistency training (Song & Dhariwal, 2024), i.e., $\boldsymbol{\theta}^- = \boldsymbol{\theta}$.

**Weight Averaging.** A line of work (Ruppert, 1988; Polyak & Juditsky, 1992; Izmailov et al., 2018) explores the integration of a running average of the weights in the context of convex optimization and stochastic gradient descent (SGD). More recently, Wortsman et al. (2022) demonstrate the potential of averaging models that have been fine-tuned with various hyperparameter configurations. In early DM studies, the use of Exponential Moving Average (EMA) is found to significantly enhance the

quality of generation. This empirical strategy has since been adopted in most, if not all, subsequent research endeavors. Consequently, CM have also incorporated this technique, discovering that EMA models perform substantially better. EMA employs a specific averaging form that uses the exponential rate $\gamma$: $\tilde{\boldsymbol{\theta}}_n = \gamma\tilde{\boldsymbol{\theta}}_{n-1} + (1-\gamma)\boldsymbol{\theta}_n$, where $n = 1, \ldots, N$ denotes the number of training iterations, $\tilde{\boldsymbol{\theta}}$ represents the EMA model, and is initialized with $\tilde{\boldsymbol{\theta}}_0 = \boldsymbol{\theta}_0$. For Large Language Models (LLMs), many works have proposed more advanced weight averaging strategies aiming at merging models fine-tuned for different downstream tasks to create a new model with multiple capabilities (Ilharco et al., 2022; Yadav et al., 2024; Jin et al., 2022; Yu et al., 2023).

Unlike these approaches, our work focuses on accelerating model convergence and achieving better performance *within a standalone training process*. Additionally, our methodology for determining merging coefficients is novel, thereby distinguishing our method from these related works.

**Search-based Methods for Diffusion Models.** Recently, many studies have proposed discrete optimization dimensions for DMs and employed search methods to discover optimal solutions. This includes using search methods to optimize model schedule for DMs (Liu et al., 2023a; Li et al., 2023; Yang et al., 2023), appropriate strategies for diffusion solvers (Liu et al., 2023b), or quantization settings for DMs (Zhao et al., 2024b;a). However, these works do not involve modification to model weights and are only applicable to DMs.

## 3    THEORETICAL AND EMPIRICAL MOTIVATION OF LCSC

In this section, we first conduct a theoretical analysis to understand the effectiveness of EMA in Sec. 3.1. A key insight from this analysis is that the optimal EMA rate is not fixed but depends on the number of training iterations, raising concerns about how to tune EMA effectively. We then present empirical evidence in Sec. 3.2, demonstrating that EMA is not special in its effectiveness. In fact, many linear combination results yield better performance on the trained models. Specifically, based on the empirical evidence, we provide further theoretical analysis to prove that EMA is a suboptimal method. These observations motivate the development of a more flexible merging method.

### 3.1    THEORETICAL CONVERGENCE ANALYSIS

**Analysis Framework.** Previous works have established various frameworks for analyzing DMs and CMs while considering their respective objectives (Sohl-Dickstein et al., 2015; Ho et al., 2020; Song et al., 2020a; Karras et al., 2022; Song et al., 2023). In this section, we follow the general optimization analysis, which represents the essential form of network optimization (Rakhlin et al., 2011; Shamir & Zhang, 2013; Harvey et al., 2019). Specifically, we denote $f(\boldsymbol{\theta}_n)$ as the computed loss of the network parameters at $n$-th iteration $\boldsymbol{\theta}_n$. For each training iteration $n = 1, \ldots, N$, we obtain an unbiased random estimate $\hat{\boldsymbol{g}}_n$ of the gradient $\nabla f(\boldsymbol{\theta}_n)$, such that $\mathbb{E}[\hat{\boldsymbol{g}}_n] = \nabla f(\boldsymbol{\theta}_n)$. Additionally, we denote $\mathbb{E}[\|\hat{\boldsymbol{g}}_n\|^2] = \boldsymbol{\sigma}_n^2$ and assume that the expected square sum is upper bounded, i.e. $\forall n, \mathbb{E}[\|\hat{\boldsymbol{g}}_n\|^2] \leq G^2$. We emphasize that due to the high-variance characteristic of DM and CM (see App. F.4 for a more detailed discussion), $\boldsymbol{\sigma}_n^2$ does not diminish during training and may be significant, potentially distinguishing the training of DM and CM from other tasks. To conduct the analysis, we assume $f$ is $\beta$-strongly convex (cf. Eq. (9)). Moreover, we consider $f$ as a non-smooth function, since natural images often exhibit features like edges and textures that lead to abrupt changes and discontinuities in their distribution. Proofs of the following analyses are provided in App. A.

Based on the above analytical framework, Shamir & Zhang (2013) prove the following theorem and show that the model of the last training iteration has a convergence rate of $O(\log(N)/N)$.

**Theorem 3.1 (Shamir & Zhang (2013))** *Suppose $f$ is $\beta$-strongly convex, and that $\mathbb{E}[\|\hat{\boldsymbol{g}}_n\|^2] \leq G^2$ for all $n = 1, \ldots, N$. Consider SGD with step sizes $\eta_n = 1/\beta n$, then for any $N > 1$, it holds that: $\mathbb{E}[f(\boldsymbol{\theta}_N) - f(\boldsymbol{\theta}^*)] \leq \frac{17G^2(1+\log(N))}{\beta N}$, where $\boldsymbol{\theta}^*$ denotes the optimal model.*

The $O(\log(N)/N)$ convergence rate of the last-iter model is proven to be tight, as a lower bound of $\Omega(\log(N)/N)$ is found by Rakhlin et al. (2011). Next, we investigate the impact of merging the historical weights using the form: $\bar{\boldsymbol{\theta}}_N^{\boldsymbol{\alpha}} = \frac{1}{A}\sum_{n=1}^{N}\alpha_n\boldsymbol{\theta}_n$, where $\boldsymbol{\alpha} = \{\alpha_1, \ldots, \alpha_N\}$ are the merging coefficients, and $A = \sum_{n=1}^{N}\alpha_n$ normalizes the sum of coefficients to 1. When EMA is applied, the coefficients can be represented as $\alpha_n = \gamma^{N-n}$, where $\gamma$ is the exponential rate.[*] We prove that EMA reduces significant terms in the convergence bound to $O(1/N)$ for large $N$.

---

[*]This formulation differs slightly from the existing EMA implementation regarding the treatment of the initial model. However, it provides mathematical convenience in the analysis. Our experimental results also show that they achieve the same performance, as the discrepancy in their coefficients is negligible.

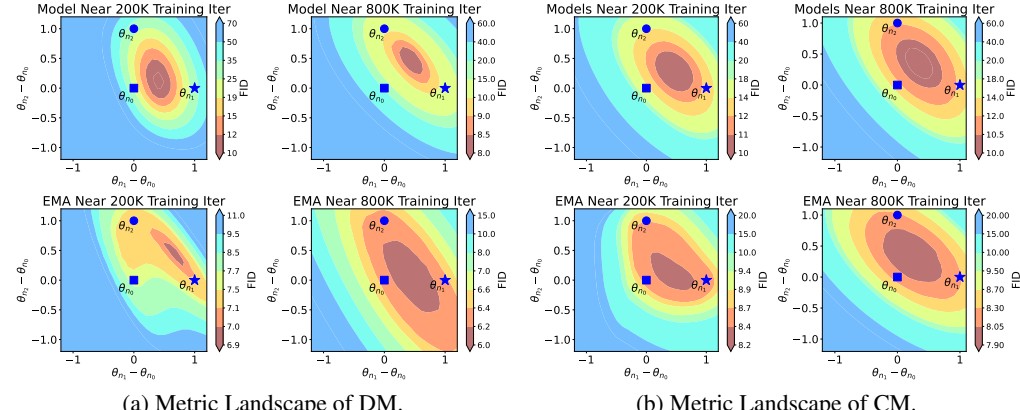

Figure 4: The metric landscape of DM and CM. Selected checkpoints $\theta_{n_0}$, $\theta_{n_1}$, and $\theta_{n_2}$ are aligned sequentially along the training trajectory, with $n_0 < n_1 < n_2$ denoting the progression in the number of training iterations. The origin point $(0,0)$ corresponds to the checkpoint $\theta_{n_1}$, while the X and Y axes quantify the differences between $\theta_{n_1} - \theta_{n_0}$ and $\theta_{n_2} - \theta_{n_0}$, respectively. A weight located at coordinate $(x,y)$ is formulated as $\theta_{(x,y)} = \theta_{n_0} + x(\theta_{n_1} - \theta_{n_0}) + y(\theta_{n_2} - \theta_{n_0})$. Additional visualizations are provided in App. D.

**Theorem 3.2** *Suppose $f$ is $\beta$-strongly convex, and that $\mathbb{E}[\|\hat{g}_n\|^2] \leq G^2$ for all $n$. Consider SGD with step sizes $\eta_n = 1/\beta n$ and EMA with factor $\gamma \in (0,1)$. Then for any $N > \lceil -\frac{1}{\ln\sqrt{\gamma}} \rceil$, we have:*

$$\mathbb{E}[f(\bar{\boldsymbol{\theta}}_N^{\boldsymbol{\alpha}}) - f(\boldsymbol{\theta}^*)] \leq \frac{G^2}{\beta} \left( \frac{1}{\gamma(1 - \gamma^{N-1})(N+1)} + \frac{v(\gamma)}{2(1 - \gamma^{N-1})} \gamma^N + 1 - \gamma \right), \quad (5)$$

*where $v(\gamma) = \sum_{j=1}^{\lceil -\frac{1}{\ln\sqrt{\gamma}} \rceil - 1} \frac{1-\gamma}{\gamma^j j} - \frac{1-\gamma}{\gamma^{\lceil -\frac{1}{\ln\sqrt{\gamma}} \rceil}}$ is a constant given $\gamma$ and independent from $N$.*

We observe that the first term in Thm. 3.2 decays as $O(1/N)$, while the second term decays exponentially as $O(\gamma^N)$. Additionally, there is a residual of $1 - \gamma$. Considering $\gamma$ is close to 1, we conclude that the significant terms (the first two terms) decay at least as fast as $O(1/N)$ and the residual term is small. We hypothesize that the improvement of these significant terms leads to an acceleration in convergence compared to the overall $O(\log(N)/N)$ rate of the last-iter model. Furthermore, we note that for sufficiently large $N$, it is preferable for $\gamma$ to be large so that the residual approaches zero, while the enlarged factors in the first two terms can be compensated by a large $N$. This insight aligns with practical experience. For example, Karras et al. (2022; 2024) implemented EMA rates that increase with the number of training iterations. Notably, the optimal EMA rate is difficult to predefine based on Thm. 3.2 without having the exact optimization landscape and the least upper bound. This motivates the search for EMA rates during or after the training. Moreover, in the next section, we will further show that EMA is suboptimal of merging coefficients per se.

## 3.2 The Effectiveness of Weight Merging and Suboptimality of EMA

To visualize the impact of linear combination of weights on the generation quality, we select 3 checkpoints $\theta_{n_0}, \theta_{n_1}, \theta_{n_2}$ from the same training trajectory. Then we sweep across the 2D space spanned by these three points and assess the FID scores. The results are shown in the first row of Fig. 4. As we can see, there exists substantial opportunities to enhance the model's performance at any training phase through the linear combination of existing weight checkpoints, which proves that the effectiveness of EMA is not special. Furthermore, we conduct additional experiments using EMA weights (see the seconde row of Fig. 4). Notably, we observe that linear combinations of the three EMA weights could also achieve superior performance. It is important to highlight that the linear combination of three EMA weights at various training iterations cannot be replicated by any single EMA weight according to Thm. 3.3. Consequently, this suggests that EMA is indeed a suboptimal solution, indicating possibilities for further improvement in this area.

**Theorem 3.3** *Assume $\boldsymbol{\theta}_1, \ldots, \boldsymbol{\theta}_N$ are linearly independent.[†] Denote $\boldsymbol{\theta}_1^\gamma, \ldots, \boldsymbol{\theta}_N^\gamma$ as the results of EMA using the exponential rate $\gamma \in (0,1)$. and select three indices $n_1, n_2, n_3$ s.t. $1 < n_1 < n_2 < n_3$. Consider three arbitrary merging coefficients $c_1, c_2, c_3 \in (0,1)$, s.t. $c_1 + c_2 + c_3 = 1$, we have:*

$$\nexists \gamma' \in (0,1) : \quad c_1 \boldsymbol{\theta}_{n_1}^\gamma + c_2 \boldsymbol{\theta}_{n_2}^\gamma + c_3 \boldsymbol{\theta}_{n_3}^\gamma \in \{\boldsymbol{\theta}_1^{\gamma'}, \ldots, \boldsymbol{\theta}_N^{\gamma'}\}. \quad (6)$$

---

[†]This assumption likely holds, as $N$ is often substantially smaller than the number of parameters.

Based on the above insights, we conclude that the optimal weight merging coefficients are difficult to predefined, and the performance of merged models can potentially be further improved with better merging coefficients. Therefore, we propose LCSC, which introduces an optimization method to search for the optimal merging coefficients.

## 4 METHOD

Next, we present the details of LCSC. The search problem is defined in Sec. 4.1. In Sec. 4.2, we elaborate on our algorithm. Finally, we list several typical use cases of our method in Sec. 4.3.

### 4.1 DEFINITION OF THE SEARCH PROBLEM

Let $\{1, 2, \ldots, N\}$ denote the set of training iterations. Suppose $\{n_1, n_2, \ldots, n_K\}$ is a subsequence selected from $\{1, 2, \ldots, N\}$. Define $\Theta$ as the set of checkpoints saved at these specific training iterations: $\Theta = \{\boldsymbol{\theta}_{n_1}, \boldsymbol{\theta}_{n_2}, \boldsymbol{\theta}_{n_3}, \cdots, \boldsymbol{\theta}_{n_K}\}$. Given $\Theta$, we aim to find a group of coefficients $\boldsymbol{\alpha} = \{\alpha_1, \alpha_2, \alpha_3, \cdots, \alpha_K\}$, that achieves the best utility when linearly combine all checkpoints:

$$\underset{\alpha_1, \alpha_2, \alpha_3, \cdots, \alpha_K \in \mathbb{R}}{\arg\min} \mathcal{F}(\alpha_1 \boldsymbol{\theta}_{n_1} + \alpha_2 \boldsymbol{\theta}_{n_2} + \alpha_3 \boldsymbol{\theta}_{n_3} + \cdots + \alpha_K \boldsymbol{\theta}_{n_K}); \quad \text{s.t.} \quad \sum_{i=1}^{K} \alpha_i = 1, \quad (7)$$

where $\mathcal{F}(\boldsymbol{\theta})$ denotes an evaluation function that measures the generation quality of $\boldsymbol{\theta}$ regarding a specific metric, indicating higher quality with a smaller value. The constraint $\sum_{i=1}^{K} \alpha_i = 1$ reduces the dimension of the search space by 1. However, we find it enables a more effective exploration for the search algorithm. Additionally, since the basin area in Fig. 4 extends beyond the convex hull of the model weights, we allow $\alpha_i < 0$, distinguishing LCSC from existing weight averaging methods.

The trained model can also be a Low Rank Adapter (LoRA) (Hu et al., 2021). In this case, each checkpoint is a product of two low-rank matrices: $\boldsymbol{\theta}_{n_i} = \boldsymbol{B}_{n_i} \boldsymbol{A}_{n_i}$. The weighted sum of all checkpoints by the coefficients is: $\alpha_1 \boldsymbol{B}_{n_1} \boldsymbol{A}_{n_1} + \alpha_2 \boldsymbol{B}_{n_2} \boldsymbol{A}_{n_2} + \alpha_3 \boldsymbol{B}_{n_3} \boldsymbol{A}_{n_3} + \cdots + \alpha_K \boldsymbol{B}_{n_K} \boldsymbol{A}_{n_K}$.

### 4.2 EVOLUTIONARY SEARCH

As discussed in Sec. 3, the optimal coefficients are difficult to predefine. To solve the problem in Sec. 4.1, we propose to use evolutionary search (Real et al., 2019; Liu et al., 2023a; Li et al., 2023).

**Our detailed algorithm is provided in Alg. 1 and an illustration is given in Fig. 3**. Given a set of checkpoints $\Theta$, we can apply LCSC at any training iteration $n$ along the training trajectory. The key points of our algorithm are summarized as follows: (a) For the stability during search, we subtract the first checkpoint from each subsequent checkpoint and search for the coefficients of their difference. (b) We initialize the population using EMA coefficients of several different rates. (c) We then conduct an evolutionary search which views a group of coefficients as an individual. In each search iteration, only the top-performing individuals are selected as parents to reproduce the next generation through crossover and mutation. Specifically, during crossover, we randomly mix the coefficients of the two parents or select one set entirely. For mutation, we introduce random Gaussian noise to each coefficient. This stochastic element ensures that beneficial adaptations are conserved and advanced to the subsequent generation, whereas detrimental modifications are discarded. (d) We repeat this reproduction process for a predetermined number of times within each search iteration and update the whole population with all newly generated individuals. (e) Upon the completion of the search process, we choose the best individual as the output of our search algorithm.

Since evolutionary search only performs model inference and is thus gradient-free, LCSC achieves significant savings in GPU memory usage (applicable to both DM and CM) and computational time (particularly for CM, given that the inference process for DM is still resource-intensive). Furthermore, LCSC has the unique advantage of optimizing models directly for non-differentiable metrics. The applicability of LCSC could potentially extends to various other tasks and models as well.

### 4.3 USE CASES

Finally, we delve into the use cases of LCSC. **(a) Decrease Training Cost.** Through our investigation, we have discerned that the training process for DM and CM can be categorized into two distinct phases, with the second phase taking the majority of training iterations but converging slowly (further discussion is deferred to App. E.2). *Our method can be employed at the onset of the second phase and yield performance on par with or even surpasses the final converged model.* Additionally, SOTA DMs and CMs necessitate training with large batch sizes, which places a substantial demand on computational resources (Karras et al., 2022; Dhariwal & Nichol, 2021; Song et al., 2023). Decreasing the batch size often leads to worse convergence speed. *LCSC can be used for low batch*

*size training and achieve equivalent performance to models trained with full batch size*. These applications aims to expedite the training process. **(b) Enhance Converged Models.** LCSC can be applied to the checkpoints saved during the final stage of training to refine the converged model. *It can enhance the generation quality under the same NFE, or maintain the generation quality with fewer NFE, thereby reducing the inference cost during deployment.* Additionally, for users who only have access to released pre-trained models, *we suggest fine-tuning the model for a few iterations and saving the intermediate checkpoints. Then LCSC can be used to enhance the released model.*

## 5 EXPERIMENTS

In Sec. 5.1, we introduce the experimental settings. In Sec. 5.2 and 5.3, we demonstrate results for the two use cases. In Sec. 5.4 we demonstrate the potential of merging LoRA checkpoints. We further apply our method on text-to-image task in Sec. 5.5. Finally, we discuss the generalization ability of LCSC in Sec. 5.6. We ablate several important hyper-parameters in our workflow in App. E.4.

### 5.1 EXPERIMENTAL SETUP

We follow previous work to configure the training process. Details of the training and search configurations are provided in App. E.1.

**Evaluation.** We choose the most commonly used metric, FID (Heusel et al., 2017), as the search objective. To evaluate the combination coefficients during the evolutionary search, we sample 5K images for ImageNet-64 with DM and LSUN datasets, while 10K for other settings, using a fixed group of initial noise. For the final model evaluation, we generate 50K samples using a different group of initial noises. To demonstrate that LCSC searches based on FID but achieves overall improvement, we also report PickScore (Kirstain et al., 2023), ImageReward (Xu et al., 2023), and the winning rates based on these two metrics for the text-to-image task, and IS (Salimans et al., 2016), precision, and recall (Kynkäänniemi et al., 2019) for other cases. In Sec. 5.6, we further report FCD (the variant of FID using CLIP features instead of Inception features) and KID (Bińkowski et al., 2018) to show that LCSC also brings improvements in other feature spaces or metrics.

**Baselines.** Since EMA is the default setting in almost all works of DM and CM, we report the performance of the EMA weights for full-model training using the rate reported by official papers (Song et al., 2023; 2020a; Nichol & Dhariwal, 2021) as our main baselines. For CM models on ImageNet-64 and LSUN datasets, we additionally download the official models trained with full batch size and test their performance for a complete comparison. We further conduct a grid search of EMA rate for the final model as a stronger baseline, denoted as EMA*, to prove the sub-optimality of EMA and that LCSC is a more effective method. More experimental details are provided in App. E.

### 5.2 RESULTS ON REDUCING THE TRAINING COST

We conduct a series of experiments on CD and CT to show that LCSC can significantly reduce their training costs. *The efficiency of LCSC has considered both training and search cost on the CPU and GPU, with detailed information available in App. E.3.* Generated images are visualized in App. H.

LCSC can be applied at an early training stage to reduce the number of training iterations. As illustrated in Tab. 1, applying LCSC to the CD model with only 50K training iterations achieves a better FID than the final model trained with 800K iterations on CIFAR-10 (3.10 vs. 3.66), which accelerates the training by approximately $14\times$. For CT, we only achieve a converged FID of 9.87 with vanilla training and are unable to fully reproduce the result of FID=8.70 reported by Song et al. (2023). However, by applying our method at 400K training iterations, we can achieve a similar FID to the reported one from 800K training iterations, achieving around a $1.9\times$ speedup. On ImageNet-64 (Tab. 2), we decrease the batch size from 2048 to 256 due to limited resources. For CD, LCSC achieves better performance (5.51 vs. 7.30 in FID) than the final converged model with only half of the training iterations (300K vs. 600K). For CT, LCSC also significantly reduces the final converged FID (10.5 vs. 15.6) with fewer training iterations (600K vs. 1000K).

LCSC can also handle smaller training batch sizes to achieve higher speedups. As illustrated in Tab. 1, LCSC with a batch size of 128 outperforms the final converged model with a batch size of 512 for both CD and CT (3.21 vs. 3.66 with CD and 8.54 vs. 9.87 with CT), achieving overall speedups of $23\times$ and $7\times$, respectively. On ImageNet-64, we test the official models of Song et al. (2023), which were trained with a 2048 batch size for both CD and CT. With the assistance of LCSC,

Table 1: Generation quality of CMs on CIFAR-10. The training speedup is compared against the standard training with 800K iterations and 512 batch size. Our results that beat or match the standard training (the "released" model for FID & IS, our reproduced results for Prec. & Rec.) are underlined.

| Model | Method | Training Iter | Batch Size | NFE | FID(↓) | IS(↑) | Prec.(↑) | Rec.(↑) | Speed(↑) |
|---|---|---|---|---|---|---|---|---|---|
| CD | EMA | 200K | 512 | 1 | 4.08 | 9.18 | 0.68 | 0.56 | |
| | | 800K | 512 | 1 | 3.66 | 9.35 | 0.68 | 0.57 | |
| | | 850K | 512 | 1 | 3.65 | 9.32 | 0.68 | 0.57 | |
| | | 850K | 512 | 2 | 2.89 | 9.55 | 0.69 | 0.58 | |
| | (released) | 800K | 512 | 1 | 3.55 | 9.48 | - | - | |
| | (released) | 800K | 512 | 2 | 2.93 | 9.75 | - | - | |
| | EMA* | 800K | 512 | 1 | 3.51 | 9.37 | 0.68 | 0.57 | |
| | LCSC | 50K | 512 | 1 | 3.10 | 9.50 | 0.66 | 0.58 | ~14× |
| | | 800K | 512 | 1 | **2.44** | **9.82** | 0.67 | 0.60 | - |
| | | 800+40K | 512 | 1 | 2.50 | 9.70 | 0.68 | 0.59 | - |
| | | 100K | 128 | 1 | 3.21 | 9.48 | 0.66 | 0.58 | ~23× |
| CT | EMA | 400K | 512 | 1 | 12.1 | 8.52 | 0.67 | 0.43 | |
| | | 800K | 512 | 1 | 9.87 | 8.81 | 0.69 | 0.42 | |
| | (released) | 800K | 512 | 1 | 8.70 | 8.49 | - | - | |
| | EMA* | 800K | 512 | 1 | 9.70 | 8.81 | 0.69 | 0.42 | |
| | LCSC | 400K | 512 | 1 | 8.89 | 8.79 | 0.67 | 0.47 | ~1.9× |
| | | 800+40K | 512 | 1 | **7.05** | **9.01** | 0.70 | 0.45 | - |
| | | 450K | 128 | 1 | 8.54 | 8.66 | 0.69 | 0.44 | ~7× |

Table 2: Generation quality of consistency models on ImageNet-64. For CD, the speedup is compared against the standard training with 600K iterations and 2048 batch size. For CT, the speedup is compared against the standard training with 800K iterations and 2048 batch size. Our results that beat the standard training ("released") are underlined.

| Model | Method | Training Iter | Batch Size | NFE | FID(↓) | IS(↑) | Prec.(↑) | Rec.(↑) | Speed(↑) |
|---|---|---|---|---|---|---|---|---|---|
| CD | EMA | 300K | 256 | 1 | 7.70 | 37.0 | 0.67 | 0.62 | |
| | | 600K | 256 | 1 | 7.30 | 37.2 | 0.67 | 0.62 | |
| | | 650K | 256 | 1 | 7.17 | 37.7 | 0.67 | 0.62 | |
| | (released) | 600K | 2048 | 1 | 6.31 | 39.5 | 0.68 | 0.63 | |
| | EMA* | 600K | 256 | 1 | 7.17 | 37.7 | 0.67 | 0.62 | |
| | LCSC | 300K | 256 | 1 | 5.51 | **39.8** | 0.68 | 0.62 | ~15× |
| | | 600+20K | 256 | 1 | **5.07** | 42.5 | 0.69 | 0.62 | ~7.6× |
| CT | EMA | 600K | 256 | 1 | 16.6 | 30.6 | 0.62 | 0.54 | |
| | | 800K | 256 | 1 | 15.8 | 31.1 | 0.64 | 0.55 | |
| | | 1000K | 256 | 1 | 15.6 | 31.2 | 0.64 | 0.55 | |
| | (released) | 800K | 2048 | 1 | 13.1 | 29.2 | 0.70 | 0.47 | |
| | EMA* | 1000K | 256 | 1 | 15.6 | 31.2 | 0.64 | 0.55 | |
| | LCSC | 600K | 256 | 1 | 10.5 | 36.8 | 0.66 | 0.56 | ~11.4× |
| | | 800K | 256 | 1 | **9.02** | **38.8** | 0.68 | 0.55 | ~7.3× |

models trained with a 256 batch size consistently outperform the official models (5.51, 5.07 vs. 6.31 with CD and 9.02, 10.5 vs. 13.1 with CT), achieving overall acceleration ratios up to 15×.

## 5.3 RESULTS ON ENHANCING PRE-TRAINED MODELS

Our experiments on both DM and CM show that LCSC can significantly boost the performance of final converged models and reduce the required sampling steps.

To simulate scenarios where users have access only to pre-trained models, we fine-tune the final converged model for a few iterations and then apply LCSC. As illustrated in the 800+40K LCSC row with CD and CT in Tab. 1, the 600+20K LCSC row with CD in Tab. 2 and the 1000+20K LCSC on both datasets in Tab. 3, this approach significantly enhances the performance of the already converged model. For model developers, they can directly utilize the checkpoints close to the final iteration, as demonstrated in the 800K LCSC row with CD in Tab. 1 and 800K LCSC row with CT in Tab. 2. LCSC leads to notable improvements in all these scenarios. Notably, on CIFAR-10, LCSC even outperforms the 800K and 850K models that use 2-step sampling, achieving better performance with 1-step sampling (2.44, 2.50 FID vs. 2.93, 2.89 FID), thereby doubling the inference speed. More experiments with Stable Diffusion checkpoints can be found at App. E.8

Same phenomenon has also been observed for DM, as shown in Tab. 4: applying LCSC at the final stage of training can obtain a model that performs significantly better than the final model of EMA. Moreover, we observe that the model found by LCSC can match the performance of the best EMA

Table 3: Generation quality of models on LSUN-bedroom and LSUN-cat with consistency training (CT). We get the checkpoints by fine-tuning the released official model.

| Dataset | Method | Training Iter | Batch Size | NFE | FID(↓) | Prec.(↑) | Rec.(↑) |
|---|---|---|---|---|---|---|---|
| LSUN-Cat | EMA | 1000K (released) | 2048 | 1 | 20.8 | 0.53 | 0.46 |
| | | 1000+20K (fine-tuned) | 256 | 1 | 21.0 | 0.58 | 0.44 |
| | LCSC | 1000+20K | 256 | 1 | **17.8** | **0.63** | **0.48** |
| LSUN-Bedroom | EMA | 1000K(released) | 2048 | 1 | 16.1 | **0.60** | 0.17 |
| | | 1000+20K(fine-tuned) | 256 | 1 | 16.0 | **0.60** | 0.17 |
| | LCSC | 1000+20K | 256 | 1 | **13.5** | 0.59 | **0.37** |

| Dataset | Method | Training Iter. | NFE | FID(↓) | IS(↑) | Prec.(↑) | Rec.(↑) |
|---|---|---|---|---|---|---|---|
| CIFAR10 | EMA | 150K | 15 | 6.28 | 8.74 | 0.61 | 0.59 |
| | | 800K | 15 | 4.16 | 9.37 | 0.64 | 0.60 |
| | EMA* | 800K | 15 | 3.96 | 9.50 | 0.64 | 0.60 |
| | LCSC | 150K | 15 | 4.76 | 8.97 | 0.61 | 0.59 |
| | | 800K | 15 | **3.18** | **9.59** | 0.64 | 0.61 |
| | | 800K | 9 | 3.97 | 9.50 | 0.63 | 0.60 |
| ImageNet | EMA | 150K | 15 | 22.3 | 15.0 | 0.55 | 0.45 |
| | | 500K | 15 | 19.8 | 16.9 | 0.58 | 0.59 |
| | EMA* | 500K | 15 | 18.1 | 17.3 | 0.59 | 0.59 |
| | LCSC | 150K | 15 | 19.1 | 15.3 | 0.56 | 0.56 |
| | | 500K | 15 | **15.3** | **17.6** | 0.59 | 0.59 |
| | | 500K | 12 | 17.2 | 17.2 | 0.57 | 0.59 |

Table 4: Generation quality of diffusion models. Our results that beat the standard training are underlined.

Table 5: Single-step generation quality of LoRA models.

| Dataset | Model | Method | Training Iter | Batch Size | FID(↓) | IS (↑) | Prec.(↑) | Rec.(↑) |
|---|---|---|---|---|---|---|---|---|
| ImageNet | CD | EMA | 600K | 256 | 7.30 | 37.21 | 0.67 | 0.62 |
| | | LoRA | 600+20K (fine-tuned with LoRA) | 256 | 6.79 | 37.74 | 0.67 | 0.63 |
| | | LCSC | 600+20K | 256 | **4.21** | **43.22** | **0.68** | **0.64** |
| | CT | EMA | 800K | 256 | 15.75 | 31.08 | 0.64 | 0.55 |
| | | LoRA | 800+20K (fine-tuned with LoRA) | 256 | 15.14 | 31.59 | 0.64 | 0.56 |
| | | LCSC | 800+20K | 256 | **4.90** | **45.33** | **0.67** | **0.65** |
| LSUN-Bedroom | CT | EMA | 1000K | 2048 | 16.10 | - | **0.60** | 0.17 |
| | | LCSC | 1000+20K | 256 | **14.49** | - | **0.60** | **0.20** |

Table 6: Text-to-image generation quality of LCM LoRA. PKS and IR stand for PickScore and ImageReward. WR stands for winning rate among all generated images compared to baselines.

| Method | Training Iter | Batch Size | Search NFE | Eval NFE | FID(↓) | PKS(↑) | WR@PKS(↑) | IR(↑) | WR@IR(↑) | CLIP-Score |
|---|---|---|---|---|---|---|---|---|---|---|
| Vanilla training | 6K | 12 | - | 4 | 32.52 | 0.46 | 34% | -2.20 | 35% | 26.02 |
| LCSC | 6K | 12 | 4 | 4 | **28.30** | **0.54** | **66%** | **-2.19** | **65%** | **26.39** |
| Vanilla training | 6K | 12 | - | 2 | 43.32 | 0.46 | 33% | -2.22 | 23% | 25.16 |
| LCSC | 6K | 12 | 2 | 2 | **30.39** | **0.54** | **67%** | **-2.20** | **77%** | **26.01** |
| Vanilla training | 6K | 12 | - | 2 | 43.32 | 0.47 | 34% | -2.22 | 25% | 25.16 |
| LCSC | 6K | 12 | 4 | 2 | **33.13** | **0.53** | **66%** | **-2.20** | **75%** | **25.89** |

model with fewer NFE during inference. Specifically, our method requiring NFE=9 is on par with EMA* using NFE=15 on CIFAR-10. Similarly, our method with NFE=12 is competitive with EMA* utilizing NFE=15 on ImageNet-64. This also highlights the potential for inference speedup.

## 5.4 RESULTS WITH LoRA CHECKPOINTS

As discussed in Sec. 4.1, we can fine-tune the model with LoRA and linearly combine the LoRA checkpoints, which significantly reduces the demand for memory and storage compared to full-model LCSC. Results are reported in Tab. 5. Fine-tuning with LoRA checkpoints achieves performance comparable to full-model fine-tuning with LCSC. On ImageNet-64 dataset with CT, we surprisingly find that applying LCSC to LoRA checkpoints outperforms the full model by a substantial margin, highlighting the effectiveness of combining LCSC with LoRA.

## 5.5 RESULTS ON TEXT-TO-IMAGE TASK

To further validate the practicality of LCSC, we apply it to the training process of LCM-LoRA (Luo et al., 2023a;b). We train the model on CC12M dataset and use 1k extra image-text pair in this dataset to conduct search. We apply 4-step sampling and 2-step sampling and observe that the search results from 4-step sampling can generalize to 2-step evaluation. We test the search result using 10k data in MS-COCO dataset with FID, PickScore, and ImageReward. Since PickScore and ImageReward are both image-wise metrics, we also report the winning rate of LCSC compared to baseline for these two metrics. More details of the setting can be found in App. E.1.3. Results are shown at Tab. 6, several examples are visualized in Fig. 2 and more are provided in Fig. 16. LCSC can significantly

Table 7: Evaluation results using FID, FCD on the training dataset and KID on the test dataset. See Tab. 18 for test set FID and Tab. 17 for results on CT.

(a) Results on CIFAR-10.

| Method | Training Iter | Batch Size | FID($\downarrow$) | FCD($\downarrow$) | KID($\downarrow$) |
|--------|---------------|------------|--------|--------|--------|
| CD | 800k | 512 | 3.66 | 18.0 | 1.38e-3 |
| CD | 800k | 512 | 3.65 | 17.3 | 1.39e-3 |
| LCSC | 100k | 128 | 3.21 | 23.4 | 7.88e-4 |
| LCSC | 250k | 512 | 2.66 | 14.3 | 4.43e-4 |
| LCSC | 800k | 512 | **2.44** | 14.7 | 4.32e-4 |
| LCSC | 800+40k | 512 | 2.50 | **14.0** | **4.11e-4** |

(b) Results on ImageNet-64.

| Method | Training Iter | Batch Size | FID($\downarrow$) | FCD($\downarrow$) | KID($\downarrow$) |
|--------|---------------|------------|--------|--------|--------|
| CD | 300k | 256 | 7.70 | 30.0 | 3.98e-3 |
| CD | 600k | 256 | 7.30 | 29.1 | 3.65e-3 |
| CD | 600k | 2048 | 6.31 | 25.5 | 3.37e-3 |
| LCSC | 300k | 256 | 5.51 | 23.9 | 3.14e-3 |
| LCSC | 800+40k | 256 | **5.07** | **23.2** | **1.97e-3** |

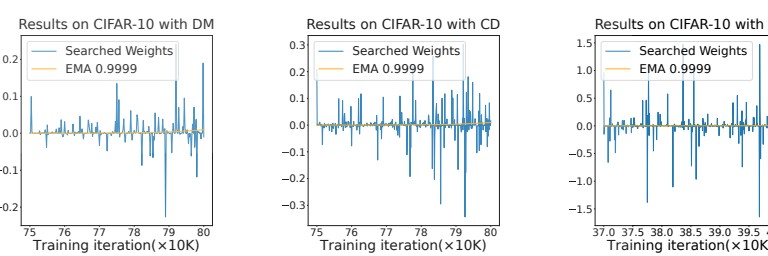

Figure 5: Visualization of weight combination coefficients obtained using LCSC and EMA.

boost the FID. For human evaluation based metrics, LCSC achieves both better average value and higher winning rate, indicating the potential of LCSC to be used in more complicated tasks.

### 5.6 DISCUSSION ON THE GENERALIZATION OF LCSC

Even if LCSC searches based on evaluation metrics (e.g., FID), we show here that LCSC generalizes across metrics and data.

**Metric Generalization.** In Sec. 5.2 to 5.5, we have evaluated the search results across various metrics besides FID, including IS, Precision, Recall, PickScore, and ImageReward. Additionally, we further evaluate the search results on the *test set* using KID (Bińkowski et al., 2018) and FCD (i.e., replacing Inception Net with CLIP as the feature extractor for FID); see Tab. 7 and Tab. 17. These metrics differ from FID in various aspects, including feature extractors (FCD, PickScore, ImageReward), calculation formulas (IS, Precision, Recall, KID, PickScore, and ImageReward), application scenarios (PickScore and ImageReward for human-based evaluation). We observe that most metrics are consistently improved, demonstrating the generalizability of LCSC across metrics.

**Data Generalization.** **(a)** *Generalization of initial noise.* During the search process, a fixed group of initial noise is used. When performing evaluation, we use a different set of noise. Therefore, the search result of LCSC generalizes to different initial noise. **(b)** *Generalization of data.* Since FID is computed using the training set as the ground truth, we calculate additional metrics based on the test set to validate that the search results of LCSC generalize across data; see Tabs. 7 and 18. Additionally, for text-to-image task (see Sec. 5.5), LCSC was searched based on CC12M but evaluated on the test dataset of MS-COCO, demonstrating generalization to a closely related data distribution.

## 6 DISCUSSION AND CONCLUSION

**Analysis of Search Patterns.** Since our method searches for the optimal coefficients, it would be interesting to check if the search pattern aligns with previous methods using fixed forms, such as EMA, and to derive insights that could inspire further research on checkpoint merging. We visualize several searched combination coefficients in Fig. 5. **(a)** First, we observe that earlier checkpoints can also be important, as some of them have large coefficients. **(b)** Moreover, we find that LCSC tend to assign large coefficients to a small subset of weights, whereas the coefficients for the majority of weights are nearly zero. Further investigation suggests smaller and homogeneous solution also exists but may be difficult to be found by LCSC (seeApp. F). **(c)** Finally, the presence of multiple significant negative coefficients highlights that certain weights can act as critical negative examples. This finding implies traditional weight-averaging methods, such as EMA, are suboptimal. Since they commonly confine the resulting model within the convex hull of all weights, which excludes the discovered solutions. More search patterns and other insights are provided in App. F.

In this work, we investigate linearly combining saved checkpoints during training to achieve better performance for DM and CM. We demonstrate the common benefits of checkpoint merging and provide a theoretical analysis to clarify that the current standard merging method is suboptimal, emphasizing the need for flexible merging coefficients. We then propose using an evolutionary algorithm to search for the optimal coefficients, which runs efficiently. Through extensive experiments, we demonstrate two uses of our method: reducing training costs and enhancing generation quality.

ACKNOWLEDGEMENT

This work was supported by National Natural Science Foundation of China (No. 62325405, 62104128, U19B2019, U21B2031, 61832007, 62204164), Flemish Government (AI Research Program) and the Research Foundation - Flanders (FWO) through project number G0G2921N, Tsinghua EE Xilinx AI Research Fund, and Beijing National Research Center for Information Science and Technology (BNRist). We thank the anonymous reviewers for their valuable feedback and suggestions. We thank Yiran Shi for his help with experiments and all the support from Infinigence-AI.

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

## A  PROOF OF THEORETICAL ANALYSIS

**Lemma A.1** *Suppose $f$ is $\beta$-strongly convex, and that $\mathbb{E}[\|\hat{g}_n\|^2] \leq G^2$ for all $n = 1, \ldots, N$. Consider SGD with step sizes $\eta_n = 1/\beta n$, then for any $N > 1$, it holds that:*

$$\mathbb{E}[\|\theta^N - \theta^*\|^2] \leq \frac{2G^2}{\beta^2 n} \tag{8}$$

Proof of Lem. A.1 basically follows Rakhlin et al. (2011), while we improve the upper bound with a factor of 2.

By the strong convexity of $f$, we have:

$$\langle g_n, \theta_n - \theta^* \rangle \geq f(\theta_n) - f(\theta^*) + \frac{\beta}{2}\|\theta_n - \theta^*\|^2, \tag{9}$$

$$f(\theta_n) - f(\theta^*) \geq \frac{\beta}{2}\|\theta_n - \theta^*\|^2. \tag{10}$$

Based on the above inequalities, we can derive:

$$\mathbb{E}[\|\boldsymbol{\theta}_{n+1} - \boldsymbol{\theta}^*\|^2] = \mathbb{E}[\|\boldsymbol{\theta}_n + \eta_n \hat{\boldsymbol{g}}_n - \boldsymbol{\theta}^*\|] \tag{11}$$

$$= \mathbb{E}[\|\boldsymbol{\theta}_n - \boldsymbol{\theta}^*\|^2] + \eta_n^2 \mathbb{E}[\|\hat{\boldsymbol{g}}_n\|^2] - 2\eta_n \mathbb{E}[\langle \boldsymbol{\theta}_n - \boldsymbol{\theta}^*, \hat{\boldsymbol{g}}_n \rangle] \tag{12}$$

$$= \mathbb{E}[\|\boldsymbol{\theta}_n - \boldsymbol{\theta}^*\|^2] + \eta_n^2 G^2 - 2\eta_n \mathbb{E}[\langle \boldsymbol{\theta}_n - \boldsymbol{\theta}^*, \boldsymbol{g}_n \rangle] \tag{13}$$

$$\overset{9}{\leq} \mathbb{E}[\|\boldsymbol{\theta}_n - \boldsymbol{\theta}^*\|^2] + \eta_n^2 G^2 - 2\eta_n (f(\boldsymbol{\theta}_n) - f(\boldsymbol{\theta}^*) + \frac{\beta}{2}\|\boldsymbol{\theta}_n - \boldsymbol{\theta}^*\|^2) \tag{14}$$

$$\overset{10}{\leq} \mathbb{E}[\|\boldsymbol{\theta}_n - \boldsymbol{\theta}^*\|^2] + \eta_n^2 G^2 - 2\eta_n \beta (\|\boldsymbol{\theta}_n - \boldsymbol{\theta}^*\|^2). \tag{15}$$

Plugging in $\eta_n = 1/\beta n$, we obtain:

$$\mathbb{E}[\|\boldsymbol{\theta}_{n+1} - \boldsymbol{\theta}^*\|^2] \leq (1 - \frac{2}{n})\mathbb{E}[\|\boldsymbol{\theta}_n - \boldsymbol{\theta}^*\|^2] + \frac{G^2}{\beta^2 n^2}. \tag{16}$$

When $n = 1$, we have:

$$\mathbb{E}[\|\boldsymbol{\theta}_2 - \boldsymbol{\theta}^*\|^2] \leq -\mathbb{E}[\|\boldsymbol{\theta}_1 - \boldsymbol{\theta}^*\|^2] + \frac{G^2}{\beta^2} \leq \frac{G^2}{\beta^2}. \tag{17}$$

Since $\mathbb{E}[\|\boldsymbol{\theta}_2 - \boldsymbol{\theta}^*\|^2] \geq 0$, we can also obtain:

$$\mathbb{E}[\|\boldsymbol{\theta}_1 - \boldsymbol{\theta}^*\|^2] \leq \frac{G^2}{\beta^2}. \tag{18}$$

Therefore, when $n = 1, 2$, Lem. A.1 is satisfied. For $n \geq 2$, we can further prove it using induction agreement:

$$\mathbb{E}[\|\boldsymbol{\theta}_{n+1} - \boldsymbol{\theta}^*\|^2] \leq (1 - \frac{2}{n})\mathbb{E}[\|\boldsymbol{\theta}_n - \boldsymbol{\theta}^*\|^2] + \frac{G^2}{\beta^2 n^2} \tag{19}$$

$$\leq (1 - \frac{2}{n})\frac{2G^2}{\beta^2 n} + \frac{G^2}{\beta^2 n^2} \tag{20}$$

$$\leq \frac{(2n-3)G^2}{\beta^2 n^2} \tag{21}$$

$$\leq \frac{2G^2}{\beta^2 (n+1)} \tag{22}$$

## A.1 PROOF OF THM. 3.2

Rearrange Eq. (12):

$$\mathbb{E}[\langle \boldsymbol{\theta}_n - \boldsymbol{\theta}^*, \boldsymbol{g}_n \rangle] = \frac{1}{2\eta_n}(\mathbb{E}[\|\boldsymbol{\theta}_n - \boldsymbol{\theta}^*\|^2] - \mathbb{E}[\|\boldsymbol{\theta}_{n+1} - \boldsymbol{\theta}^*\|^2]) + \frac{\eta_n}{2}\mathbb{E}[\|\hat{\boldsymbol{g}}_n\|^2]. \tag{23}$$

Plugging in Eq. (9) and $\eta_n = 1/\beta n$, we obtain:

$$\mathbb{E}[f(\boldsymbol{\theta}_n) - f(\boldsymbol{\theta}^*)] \leq \frac{\beta}{2}\left((n-1)\mathbb{E}[\|\boldsymbol{\theta}_n - \boldsymbol{\theta}^*\|^2] - n\mathbb{E}[\|\boldsymbol{\theta}_{n+1} - \boldsymbol{\theta}^*\|^2]\right) + \frac{1}{2\beta n}G^2. \tag{24}$$

Summing over $N$ iterations using coefficients $\boldsymbol{\alpha}$:

$$\mathbb{E}[f(\bar{\boldsymbol{\theta}}_n^{\boldsymbol{\alpha}}) - f(\boldsymbol{\theta}^*)] \tag{25}$$

$$\leq \frac{\beta}{2\mathrm{A}}\sum_{n=1}^{N}\alpha_n\left((n-1)\mathbb{E}[\|\boldsymbol{\theta}_n - \boldsymbol{\theta}^*\|^2] - n\mathbb{E}[\|\boldsymbol{\theta}_{n+1} - \boldsymbol{\theta}^*\|^2]\right) + \frac{G^2}{2\beta\mathrm{A}}\sum_{n=1}^{N}\frac{\alpha_n}{n} \tag{26}$$

$$= \frac{\beta}{2\mathrm{A}}\left(\sum_{n=2}^{N}(\alpha_n - \alpha_{n-1})(n-1)\mathbb{E}[\|\boldsymbol{\theta}_n - \boldsymbol{\theta}^*\|^2] - N\mathbb{E}[\|\boldsymbol{\theta}_{N+1} - \boldsymbol{\theta}^*\|^2]\right) + \frac{G^2}{2\beta\mathrm{A}}\sum_{n=1}^{N}\frac{\alpha_n}{n} \tag{27}$$

$$\leq \frac{\beta}{2\mathrm{A}}\sum_{n=2}^{N}(\alpha_n - \alpha_{n-1})(n-1)\mathbb{E}[\|\boldsymbol{\theta}_n - \boldsymbol{\theta}^*\|^2] + \frac{G^2}{2\beta\mathrm{A}}\sum_{n=1}^{N}\frac{\alpha_n}{n}. \tag{28}$$

Plugging in Lem. A.1, we have:

$$\mathbb{E}[f(\bar{\boldsymbol{\theta}}_n^{\boldsymbol{\alpha}}) - f(\boldsymbol{\theta}^*)] \le \frac{G^2}{\beta A} \sum_{n=2}^{N} \frac{(\alpha_n - \alpha_{n-1})(n-1)}{n} + \frac{G^2}{2\beta A} \sum_{n=1}^{N} \frac{\alpha_n}{n} \tag{29}$$

$$\le \frac{G^2}{\beta A} \sum_{n=2}^{N} (\alpha_n - \alpha_{n-1}) + \frac{G^2}{2\beta A} \sum_{n=1}^{N} \frac{\alpha_n}{n}. \tag{30}$$

Plugging in $\alpha_n = \gamma^{N-n}$, $A = \frac{1-\gamma^{N-1}}{1-\gamma}$, we have

$$\mathbb{E}[f(\bar{\boldsymbol{\theta}}_n^{\boldsymbol{\alpha}}) - f(\boldsymbol{\theta}^*)] \le \frac{G^2}{\beta A} \sum_{n=2}^{N} \left( \gamma^{N-n} - \gamma^{N-n+1} \right) + \frac{G^2}{2\beta A} \sum_{n=1}^{N} \frac{\gamma^{N-n}}{n} \tag{31}$$

$$= \frac{G^2(1-\gamma)}{\beta A} \sum_{n=2}^{N} \gamma^{N-n} + \frac{G^2}{2\beta A} \gamma^N \sum_{n=1}^{N} \frac{1}{\gamma^n n} \tag{32}$$

$$= \frac{G^2(1-\gamma)}{\beta A} \frac{1-\gamma^{N-2}}{1-\gamma} + \frac{G^2}{2\beta A} \gamma^N \sum_{n=1}^{N} \frac{1}{\gamma^n n} \tag{33}$$

$$= \frac{G^2(1-\gamma)}{\beta} \frac{1-\gamma^{N-2}}{1-\gamma^{N-1}} + \frac{G^2(1-\gamma)}{2\beta(1-\gamma^{N-1})} \gamma^N \sum_{n=1}^{N} \frac{1}{\gamma^n n} \tag{34}$$

$$\le \frac{G^2(1-\gamma)}{\beta} + \frac{G^2(1-\gamma)}{2\beta(1-\gamma^{N-1})} \gamma^N \sum_{n=1}^{N} \frac{1}{\gamma^n n}. \tag{35}$$

Next, we focus on deriving an upper bound for the second term in Eq. (35), since there is no simple closed-form expression for it. We notice that $\gamma^n$ decays faster than $n$ grows. Therefore it is more important to evaluate $\frac{1}{\gamma^n n}$ when $n$ approaches $N$.

We will slightly abuse the notation $x$ to denote a positive real number. First, we note that:

$$d\frac{1}{\gamma^x x} = -(\gamma^x x)^{-2}(\gamma^x + x\gamma^x \ln \gamma) \tag{36}$$

$$= -\frac{1/x + \ln \gamma}{\gamma^x x}. \tag{37}$$

If $x \ge -\frac{1}{\ln \sqrt{\gamma}}$, we have $1/x \le \ln 1/\sqrt{\gamma}$. Plug this into Eq. (37):

$$if\ x \ge -\frac{1}{\ln \sqrt{\gamma}}, \quad d\frac{1}{\gamma^x x} \ge -\frac{\ln 1/\sqrt{\gamma} + \ln \gamma}{\gamma^x x} \tag{38}$$

$$\ge \frac{-\ln \sqrt{\gamma}}{\gamma^x x}. \tag{39}$$

Integrating both sides from $x = \lceil -\frac{1}{\ln \sqrt{\gamma}} \rceil$ to $x = N + 1$, we obtain:

$$\int_{x=\lceil -\frac{1}{\ln \sqrt{\gamma}} \rceil}^{x=N+1} d\frac{1}{\gamma^x x} = \frac{1}{\gamma^{N+1}(N+1)} - \frac{1}{\gamma^{\lceil -\frac{1}{\ln \sqrt{\gamma}} \rceil} \lceil -\frac{1}{\ln \sqrt{\gamma}} \rceil} \tag{40}$$

$$\overset{39}{\ge} \int_{x=\lceil -\frac{1}{\ln \sqrt{\gamma}} \rceil}^{x=N+1} \frac{-\ln \sqrt{\gamma}}{\gamma^x x} dx. \tag{41}$$

Since the derivative in Eq. (39) is always positive, the function $\frac{1}{\gamma^x x}$ is monotonically increasing when $x \geq -1/\ln \sqrt{\gamma}$. In such case, the lower Riemann sums underestimate the integral, we have:

$$\sum_{n=\lceil -\frac{1}{\ln \sqrt{\gamma}} \rceil}^{N} \frac{1}{\gamma^n n} \leq \int_{x=\lceil -\frac{1}{\ln \sqrt{\gamma}} \rceil}^{x=N+1} \frac{1}{\gamma^x x} \mathrm{d}x \tag{42}$$

$$\overset{41}{\leq} \frac{1}{-\ln \sqrt{\gamma}} \left( \frac{1}{\gamma^{N+1}(N+1)} - \frac{1}{\gamma^{\lceil -\frac{1}{\ln \sqrt{\gamma}} \rceil} \lceil -\frac{1}{\ln \sqrt{\gamma}} \rceil} \right) \tag{43}$$

$$\leq -\frac{1}{\gamma^{N+1}(N+1) \ln \sqrt{\gamma}} - \frac{1}{\gamma^{\lceil -\frac{1}{\ln \sqrt{\gamma}} \rceil}}. \tag{44}$$

Plugging Eq. (44) into Eq. (35), we have:

$$\mathbb{E}[f(\boldsymbol{\theta}_n^\alpha) - f(\boldsymbol{\theta}*)] \leq \frac{G^2(1-\gamma)}{\beta} + \frac{G^2(1-\gamma)}{2\beta(1-\gamma^{N-1})}\gamma^N \left( \sum_{n=\lceil -\frac{1}{\ln \sqrt{\gamma}} \rceil}^{N} \frac{1}{\gamma^n n} + \sum_{j=1}^{\lceil -\frac{1}{\ln \sqrt{\gamma}} \rceil - 1} \frac{1}{\gamma^j j} \right) \tag{45}$$

$$\leq \frac{G^2(1-\gamma)}{\beta} + \frac{G^2(1-\gamma)}{2\beta(1-\gamma^{N-1})}\gamma^N \Big($$

$$- \frac{1}{\gamma^{N+1}(N+1) \ln \sqrt{\gamma}} - \frac{1}{\gamma^{\lceil -\frac{1}{\ln \sqrt{\gamma}} \rceil}} + \sum_{j=1}^{\lceil -\frac{1}{\ln \sqrt{\gamma}} \rceil - 1} \frac{1}{\gamma^j j} \Big) \tag{46}$$

$$= \frac{G^2(1-\gamma)}{\beta} + \frac{G^2}{2\beta(1-\gamma^{N-1})} \left( \frac{1-\gamma}{\gamma(N+1) \ln 1/\sqrt{\gamma}} + \gamma^N v(\gamma) \right) \tag{47}$$

$$\leq \frac{G^2(1-\gamma)}{\beta} + \frac{G^2}{2\beta(1-\gamma^{N-1})} \left( \frac{2}{\gamma(N+1)} + \gamma^N v(\gamma) \right) \tag{48}$$

$$= \frac{G^2}{\beta} \left( \frac{1}{\gamma(1-\gamma^{N-1})(N+1)} + \frac{v(\gamma)}{2(1-\gamma^{N-1})}\gamma^N + 1 - \gamma \right), \tag{49}$$

where $v(\gamma) = \sum_{j=1}^{\lceil -\frac{1}{\ln \sqrt{\gamma}} \rceil - 1} \frac{1-\gamma}{\gamma^j j} - \frac{1-\gamma}{\gamma^{\lceil -\frac{1}{\ln \sqrt{\gamma}} \rceil}}$. To derive Eq. (48), we first identify that $(1 - \gamma)/\ln 1/\sqrt{\gamma}$ is monotonically increasing for $\gamma \in (0, 1)$. We then compute the limit as $\gamma \to 1$ using L'Hôpital's rule.

## A.2  PROOF OF THM. 3.3

We prove Thm. 3.3 using a proof by contradiction.

Assume $\exists \gamma' \in (0, 1)$, s.t. $c_1 \boldsymbol{\theta}_{n_1}^\gamma + c_2 \boldsymbol{\theta}_{n_2}^\gamma + c_3 \boldsymbol{\theta}_{n_3}^\gamma \in \{\boldsymbol{\theta}_1^{\gamma'}, \ldots, \boldsymbol{\theta}_N^{\gamma'}\}$. Denote $n^* \in \{1 \ldots N\}$ such that:

$$c_1 \boldsymbol{\theta}_{n_1}^\gamma + c_2 \boldsymbol{\theta}_{n_2}^\gamma + c_3 \boldsymbol{\theta}_{n_3}^\gamma = \boldsymbol{\theta}_{n^*}^{\gamma'}. \tag{50}$$

Since $\boldsymbol{\theta}_1^\gamma, \ldots, \boldsymbol{\theta}_N^\gamma$ are the EMA of $\boldsymbol{\theta}_1, \ldots, \boldsymbol{\theta}_N$, which are linearly independent, and $\boldsymbol{\theta}_{n^*}^{\gamma'}$ is an EMA of $\boldsymbol{\theta}_1, \ldots, \boldsymbol{\theta}_{n^*}$, we can derive: $n^* = n_3$.

Substituting the EMA models with the original models, we have:

$$c_1 \boldsymbol{\theta}_{n-1}^\gamma + c_2 \boldsymbol{\theta}_{n_2}^\gamma + c_3 \boldsymbol{\theta}_{n_3}^\gamma = (c_1 \gamma^{n_1-1} + c_2 \gamma^{n_2-1} + c_3 \gamma^{n_3-1}) \boldsymbol{\theta}_1 + \ldots$$
$$+ (1-\gamma) \left( c_1 + c_2 \gamma^{n_2-n_1} + c_3 \gamma^{n_3-n_1} \right) \boldsymbol{\theta}_{n_1} + \ldots$$
$$+ (1-\gamma)(c_2 + c_3 \gamma^{n_3-n_2}) \boldsymbol{\theta}_{n_2} + \ldots + (1-\gamma)c_3 \boldsymbol{\theta}_{n_3}, \tag{51}$$
$$\boldsymbol{\theta}_{n_3}^{\gamma'} = \gamma'^{n_3-1} \boldsymbol{\theta}_1 + \ldots + (1-\gamma')\gamma'^{n_3-n_1} \boldsymbol{\theta}_{n_1} + (1-\gamma')\gamma'^{n_3-n_2} \boldsymbol{\theta}_{n_2} + \ldots$$
$$+ (1-\gamma') \boldsymbol{\theta}_{n_3}. \tag{52}$$

Note that we apply the practical implementation of EMA for Thm. 3.3, which is different from Thm. 3.2. However, their discrepancy is negligible. Since $\boldsymbol{\theta}_1^\gamma, \ldots, \boldsymbol{\theta}_N^\gamma$ are linearly independent,

coefficients of all models $\boldsymbol{\theta}_1, \ldots, \boldsymbol{\theta}_{n_3}$ in Eqs. (51) and (52) should be aligned with each other. Based on the last term in Eqs. (51) and (52), we have:

$$(1 - \gamma)c_3 = 1 - \gamma' \Rightarrow \gamma' = 1 - (1 - \gamma)c_3. \tag{53}$$

**Case 1:** $n_2 < n_3 - 1$  First, we consider the case $n_2 < n_3 - 1$, we have the following equations for the index $n_3 - 1$:

$$(1 - \gamma)\gamma c_3 \boldsymbol{\theta}_{n_3-1} = (1 - \gamma')\gamma' \boldsymbol{\theta}_{n_3-1} \tag{54}$$

$$(1 - \gamma)\gamma c_3 = (1 - \gamma')\gamma' \tag{55}$$

$$(1 - \gamma)\gamma c_3 \overset{53}{=} (1 - \gamma)c_3 \gamma' \tag{56}$$

$$\gamma = \gamma' \tag{57}$$

$$\gamma' \overset{53}{=} 1 - (1 - \gamma')c_3 \tag{58}$$

$$\gamma'(1 - c_3) = 1 - c_3 \tag{59}$$

$$\gamma' = 1. \tag{60}$$

Eq. (60) contradicts with the assumption $\gamma, \gamma' \in (0, 1)$.

**Case 2:** $n_2 = n_3 - 1$ **and** $n_1 < n_2 - 1$  Next, we consider the case $n_2 = n_3 - 1$ and $n_1 < n_2 - 1$. We have the following equation for the index $n_2$:

$$(1 - \gamma')\gamma' \boldsymbol{\theta}_{n_2} = (1 - \gamma)\gamma c_3 \boldsymbol{\theta}_{n_2} + (1 - \gamma)c_2 \boldsymbol{\theta}_{n_2}. \tag{61}$$

Additionally, we have the equations below for the index $n_2 - 1$:

$$(1 - \gamma')\gamma'^2 = (1 - \gamma)\gamma^2 c_3 + (1 - \gamma)\gamma c_2 \tag{62}$$

$$((1 - \gamma)\gamma c_3 + (1 - \gamma)c_2)\gamma' \overset{61}{=} ((1 - \gamma)\gamma c_3 + (1 - \gamma)c_2)\gamma \tag{63}$$

$$\gamma' = \gamma \tag{64}$$

$$\gamma' \overset{57-60}{=} 1. \tag{65}$$

Eq. (65) contradicts with the assumption $\gamma, \gamma' \in (0, 1)$.

**Case 3:** $n_2 = n_3 - 1$ **and** $n_1 = n_2 - 1$  Finally, we consider the case $n_2 = n_3 - 1$ and $n_1 = n_2 - 1$. We have the following equation for the index $n_1$:

$$(1 - \gamma')\gamma'^2 = (1 - \gamma)\gamma^2 c_3 + (1 - \gamma)\gamma c_2 + (1 - \gamma)c_1. \tag{66}$$

Additionally, we have the equations below for the index $n_1 - 1$:

$$(1 - \gamma')\gamma'^3 = (1 - \gamma)\gamma^3 c_3 + (1 - \gamma)\gamma^2 c_2 + (1 - \gamma)\gamma c_1 \tag{67}$$

$$\left((1 - \gamma)\gamma^2 c_3 + (1 - \gamma)\gamma c_2 + (1 - \gamma)c_1\right)\gamma' \overset{66}{=} \left((1 - \gamma)\gamma^2 c_3 + (1 - \gamma)\gamma c_2 + (1 - \gamma)c_1\right)\gamma \tag{68}$$

$$\gamma' = \gamma \tag{69}$$

$$\gamma' \overset{57-60}{=} 1. \tag{70}$$

Eq. (70) contradicts with the condition $\gamma, \gamma' \in (0, 1)$.

Since the condition $\gamma, \gamma' \in (0, 1)$ always leads to a contradiction, we prove that $\nexists \gamma' \in (0, 1)$, s.t. $c_1 \boldsymbol{\theta}_{n_1}^{\gamma} + c_2 \boldsymbol{\theta}_{n_2}^{\gamma} + c_3 \boldsymbol{\theta}_{n_3}^{\gamma} \in \{\boldsymbol{\theta}_1^{\gamma'}, \ldots, \boldsymbol{\theta}_N^{\gamma'}\}$.

## B  EXTENDED BACKGROUND AND RELATED WORK

### B.1  DIFFUSION PROBABILISTIC MODEL

Let us denote the data distribution by $p_{data}$ and consider a diffusion process that perturbs $p_{data}$ with a stochastic differential equation (SDE) (Song et al., 2020b):

$$d\mathbf{x}_t = \boldsymbol{\mu}(\mathbf{x}_t, t)dt + \sigma(t)d\boldsymbol{w}_t, \tag{71}$$

where $\boldsymbol{\mu}(\cdot,\cdot)$ and $\sigma(\cdot)$ represent the drift and diffusion coefficients, respectively, $\boldsymbol{w}_t$ denotes the standard Brownian motion, and $t \in [0, T]$ indicates the time step. $t = 0$ stands for the real data distribution. $\boldsymbol{\mu}(\cdot,\cdot)$ and $\sigma(\cdot)$ are designed to make sure $p_T(\boldsymbol{x})$ becomes pure Gaussian noise.

Diffusion models (DM) (Sohl-Dickstein et al., 2015; Song et al., 2020a; Ho et al., 2020; Nichol & Dhariwal, 2021; Song et al., 2020b; Karras et al., 2022; Dhariwal & Nichol, 2021) undertake the reverse operation by initiating with $\boldsymbol{x}_T$ sampled from pure Gaussian noise and progressively denoising it to reconstruct the image $\boldsymbol{x}_0$. Importantly, SDE has its corresponding "probability flow" Ordinary Differential Equation (PF ODE) (Song et al., 2020b;a), which delineates a deterministic pathway that yields the same distribution $p_t(\boldsymbol{x})$ for $\forall t$, thereby offering a more efficient sampling mechanism:

$$d\mathbf{x}_t = \left[ \boldsymbol{\mu}(\mathbf{x}_t, t) - \frac{1}{2}\sigma(t)^2 \nabla \log p_t(\mathbf{x}) \right] dt, \qquad (72)$$

where $\nabla \log p_t(\boldsymbol{x})$ is referred to as the *score function* of $p_t(\boldsymbol{x})$ (Hyvärinen, 2005). Various ODE solvers have been introduced to further expedite the sampling process utilizing Eq. (72) or minimizing the truncation error (Liu et al., 2022; Jolicoeur-Martineau et al., 2021; Karras et al., 2022; Lu et al., 2022).

A pivotal insight within diffusion models is the realization that $\nabla \log p_t(\boldsymbol{x})$ can be approximated by a neural network $\boldsymbol{s_\theta}(\boldsymbol{x}_t, t)$, which can be trained using the following objective:

$$\mathbb{E}_{t \sim \mathcal{U}(0,T]}\mathbb{E}_{\mathbf{y} \sim p_{\text{data}}}\mathbb{E}_{\mathbf{x}_t \sim \mathcal{N}(\mathbf{y}, \sigma(t)^2 \mathbf{I})}\lambda(t)\|\boldsymbol{s_\theta}(\mathbf{x}_t, t) - \nabla_{\mathbf{x}_t} \log p(\mathbf{x}_t|\mathbf{y})\|, \qquad (73)$$

where $\lambda(t)$ represents the loss weighting and $\boldsymbol{y}$ denotes a training image.

## B.2 CONSISTENCY MODELS

Drawing inspiration from DM theory, consistency models (CM) have been proposed to enable single-step generation (Song et al., 2023; Song & Dhariwal, 2024). Whereas DM incrementally denoises an image, e.g., via the PF ODE, CM denoted by $f_{\boldsymbol{\theta}}$ is designed to map any point $\boldsymbol{x}_t$ at any given time $t$ along a PF ODE trajectory directly to the trajectory's initial point $\boldsymbol{x}_t$ in a single step.

CMs are usually trained through discretized time steps, so we consider segmenting the time span from $[\epsilon, T]$ into $K - 1$ sub-intervals, with $\epsilon$ being a small value approximating zero. Training of CM can follow one of two primary methodologies: consistency distillation (CD) or direct consistency training (CT). In the case of CD, the model $f_{\boldsymbol{\theta}}$ leverages knowledge distilled from a pre-trained DM $\phi$. The distillation loss can be formulated as follows:

$$\mathbb{E}_{k \sim \mathcal{U}[1, K-1]}\mathbb{E}_{\boldsymbol{y} \sim p_{data}}\mathbb{E}_{\boldsymbol{x}_{t_{k+1}} \sim \mathcal{N}(\boldsymbol{y}, t_{k+1}^2 \boldsymbol{I})}\lambda(t_k)d(f_{\boldsymbol{\theta}}(\boldsymbol{x}_{t_{k+1}}, t_{k+1}), f_{\boldsymbol{\theta}^-}(\hat{\boldsymbol{x}}_{t_k}^\phi, t_k)), \qquad (74)$$

where $f_{\boldsymbol{\theta}^-}$ refers to the target model and $\boldsymbol{\theta}^-$ is computed through EMA of the historical weights of $\boldsymbol{\theta}$, $\hat{\boldsymbol{x}}_{t_k}^\phi$ is estimated by the pre-trained diffusion model $\phi$ through one-step denoising based on $\boldsymbol{x}_{t_{k+1}}$, and $d$ is a metric implemented by either the $\ell_2$ distance or LPIPS (Zhang et al., 2018).

Alternatively, in the CT case, the models $f_{\boldsymbol{\theta}}$ are developed independently, without relying on any pre-trained DM:

$$\mathbb{E}_{k \sim \mathcal{U}[1, K-1]}\mathbb{E}_{\boldsymbol{y} \sim p_{data}}\mathbb{E}_{\boldsymbol{z} \sim \mathcal{N}(\mathbf{0}, \boldsymbol{I})}\lambda(t_k)d(f_{\boldsymbol{\theta}}(\boldsymbol{y} + t_{k+1}\boldsymbol{z}, t_{k+1}), f_{\boldsymbol{\theta}^-}(\boldsymbol{y} + t_k\boldsymbol{z}, t_k)), \qquad (75)$$

where the target model $f_{\boldsymbol{\theta}^-}$ is set to be the same as the model $f_{\boldsymbol{\theta}}$ in the latest improved version of the consistency training (Song & Dhariwal, 2024), i.e. $\boldsymbol{\theta}^- = \boldsymbol{\theta}$.

## B.3 WEIGHT AVERAGING

The integration of a running average of the weights by stochastic gradient descent (SGD) was initially explored within the realm of convex optimization (Ruppert, 1988; Polyak & Juditsky, 1992). This concept was later applied to the training of neural networks (Szegedy et al., 2016). Izmailov et al.(Izmailov et al., 2018) suggest that averaging multiple weights over the course of training can yield better generalization than SGD. Wortsman et al.(Wortsman et al., 2022) demonstrate the potential of averaging weights that have been fine-tuned with various hyperparameter configurations. The Exponential Moving Average (EMA) employs a specific form that uses the exponential rate $\gamma$ as a smoothing factor:

$$\tilde{\boldsymbol{\theta}}_n = \gamma\tilde{\boldsymbol{\theta}}_{n-1} + (1 - \gamma)\boldsymbol{\theta}_n, \qquad (76)$$

where $n = 1, \ldots, N$ denotes the number of training iterations, $\tilde{\theta}$ represents the EMA model, and is initialized with $\tilde{\theta}_0 = \theta_0$.

Practitioners often opt for advanced optimizers such as Adam (Kingma & Ba, 2014) for different tasks and network architectures, which might reduce the need for employing a running average. However, the use of EMA has been noted to significantly enhance the quality of generation in early DM studies (Song et al., 2020b; Dhariwal & Nichol, 2021; Nichol & Dhariwal, 2021; Ho et al., 2020; Song et al., 2020a). This empirical strategy has been adopted in most, if not all, subsequent research endeavors. Consequently, CM have also incorporated this technique, discovering EMA models that perform substantially better.

Recently, several works of more advanced weight averaging strategies have been proposed for Large Language Models (LLMs). Most of them focus on merging models fine-tuned for different downstream tasks to create a new model with multiple capabilities (Ilharco et al., 2022; Yadav et al., 2024; Jin et al., 2022; Yu et al., 2023). Different from these approaches, our work aims to accelerate the model convergence and achieve better performance in a standalone training process. Moreover, the methodology we employ for determining averaging coefficients is novel, thereby distinguishing our method from these related works. Further details can be found in Sec. 3 and Sec. 4.

### B.4 SEARCH-BASED METHODS FOR DIFFUSION MODELS

Search algorithms are widely used across various domains like Neural Architecture Search (NAS) (Real et al., 2019; Ning et al., 2020), where they are employed to identify specific targets. Recently, many works studies have set discrete optimization dimensions for DMs and utilized search methods to unearth optimal solutions. For example, Liu et al. (Liu et al., 2023a), Li et al. (Li et al., 2023) and Yang et al. (Yang et al., 2023) use search methods to find the best model schedule for DMs. Liu et al. (Liu et al., 2023b) apply search methods to find appropriate strategies for diffusion solvers. However, these works do not involve modification to model weights during search and are only applicable to DMs.

## C LCSC ALGORITHM

The detailed algorithm of LCSC is provided in Alg. 1.

## D ADDITIONAL EXAMPLES OF THE METRIC LANDSCAPE

We previously introduced the metric landscape of the DM and CD models in Sec. 3. In this section, we extend our analysis by evaluating additional metric landscapes using the FID. Specifically, we incorporate an additional intermediate training iteration and explore the metric landscape of the CT model. The comprehensive landscapes of the DM, CD, and CT models are depicted in Fig. 7.

Since we use grid search to get the performance and using interpolation to get the smooth landscapes, we further provide some examples of the original performance heat-map in grids, as shown in Fig. 6

## E EXPERIMENTAL DETAILS AND ADDITIONAL RESULTS

### E.1 EXPERIMENTAL DETAILS

#### E.1.1 TRAINING SETTING

Since LCSC employs weight checkpoints, we endeavored to replicate the training processes of the baseline models.

For DM, we follow DDIM (Song et al., 2020a) for the evaluation on CIFAR10 (Krizhevsky et al., 2009) and iDDPM (Nichol & Dhariwal, 2021) for the evaluation on ImageNet-64 (Deng et al., 2009). To improve the inference efficiency, we adopt DPM-Solver (Lu et al., 2022). For CM (Song et al., 2023), we evaluate LCSC with both CD and CT on CIFAR-10 and ImageNet-64, and CT on LSUN datasets. For CIFAR-10 and ImageNet-64, We train models with our own implementation. We follow all the settings reported by the official paper except ImageNet-64, on which we decrease the batch size to 256 on CM and 512 on DM due to the limited resources. We apply LCSC at

---

**Algorithm 1** Evolutionary Search for Combination Coefficients Optimization

---

**Require:**

    $\Theta = \{\boldsymbol{\theta}_{n_1}, \boldsymbol{\theta}_{n_2}, \boldsymbol{\theta}_{n_3}, \cdots, \boldsymbol{\theta}_{n_K}\}$: the set of saved checkpoints untill training iter $T$

    $\mathcal{F}$: the metric evaluator.

**Symbols:**

    $P$: The whole *P*opulation of model schedule.

    $CP$: The *C*andidate *P*arents set of each loop, from which a parent coefficients is selected.

    $NG$: The *N*ext *G*eneration newly mutated from the parent coefficients in each loop.

    $\boldsymbol{\alpha}$: A group of combination coefficients denoted as $\boldsymbol{\alpha} = \{\alpha_1, \cdots, \alpha_K\}$

    $\boldsymbol{\alpha} \cdot \Theta$: Equal to $(1 - \sum_{i=2}^{K} \alpha_i)\boldsymbol{\theta}_{n_1} + \sum_{i=2}^{K} \alpha_i \boldsymbol{\theta}_{n_i}$

**Hyperparameters:**

    **Epoch**: Number of loops for the entire search process.

    $\mathbf{M}_{CP}$: Maximum size of the candidate parents set $CP$.

    **Iter**: Maximum number of mutations in each loop.

**Search Process:**

  1:  $P \leftarrow \varnothing$

  2:  Initialize a group of coefficients $\boldsymbol{\alpha}_{init}$ with EMA weights

  3:  $P \leftarrow P \cup \{(\boldsymbol{\alpha}_{init}, \mathcal{F}(\boldsymbol{\alpha}_{init} \cdot \Theta))\}$

  4:  **for** $t = 1, \cdots, $**Epoch do**

  5:     $NG \leftarrow \varnothing$

  6:     **for** $i = 1, \cdots, $**Iter do**

  7:        $CP \leftarrow \{\boldsymbol{\alpha}_i | \mathcal{F}(\boldsymbol{\alpha}_i \cdot \Theta)$ ranks within the top $min(\mathbf{M}_{CP}, |P|)$ in $P\}$

  8:        $\boldsymbol{\alpha}_f, \boldsymbol{\alpha}_m \xleftarrow{\text{Random Sample}} CP$

  9:        $\boldsymbol{\alpha}_{new} \leftarrow \text{Mutate}(\text{Crossover}(\boldsymbol{\alpha}_f, \boldsymbol{\alpha}_m))$

10:        $NG \leftarrow NG \cup \{(\boldsymbol{\alpha}_{new}, \mathcal{F}(\boldsymbol{\alpha}_{new} \cdot \Theta))\}$

11:     **end for**

12:     $P \leftarrow P \cup NG$

13:  **end for**

14:  $\boldsymbol{\alpha}^* \leftarrow \arg\min_{\boldsymbol{\alpha} \in P} \mathcal{F}(\boldsymbol{\alpha} \cdot \Theta)$

15:  **return** $\boldsymbol{\alpha}^* \cdot \Theta$

---

different training stages. For LSUN datasets, we fine-tune the released official models and apply LCSC. We further train a LCM-LoRA model following the official setting and use LCSC with the final checkpoints. Specifically, for any selected iteration, we utilize checkpoints from every predetermined interval of iterations within a defined window size. Then we run a search process to find the optimal combination coefficients.

For DMs, we were able to reproduce the results reported in the original papers successfully. However, for CMs, our training outcomes on CIFAR10 were slightly inferior to those documented in the original papers. Additionally, due to resource constraints, we opted for a smaller batch size than the original configuration when training on ImageNet. In the results tables, we present both our training outcomes and the results reported in the original papers, with the latter indicated as *released*. The specifics of the experimental setups are detailed below.

For the experiments with DM, we utilize the DDIM (Song et al., 2020a) codebase (`https://github.com/ermongroup/ddim`) on CIFAR-10, adhering to the default configuration settings. For ImageNet-64, we employ the iDDPM (Nichol & Dhariwal, 2021) codebase (`https://github.com/openai/improved-diffusion`), setting the batch size to 512, the noise schedule to cosine, and maintaining other hyperparameters at their default values. During sampling, the DPM-Solver (Lu et al., 2022) (`https://github.com/LuChengTHU/dpm-solver`) is applied, conducting 15 timesteps of denoising using the default configuration for the respective setting. The reproduced results are on par with or slightly surpass those achieved using Euler integration, as reported in the original papers.

For the experiments involving CM, we adhere closely to the configurations detailed in Table 3 of the original CM paper (Song et al., 2023). On CIFAR-10, instead of using the NCSNPP model (Song et al., 2020b) from CM's official implementation, we opt for the EDM (Karras et al.,

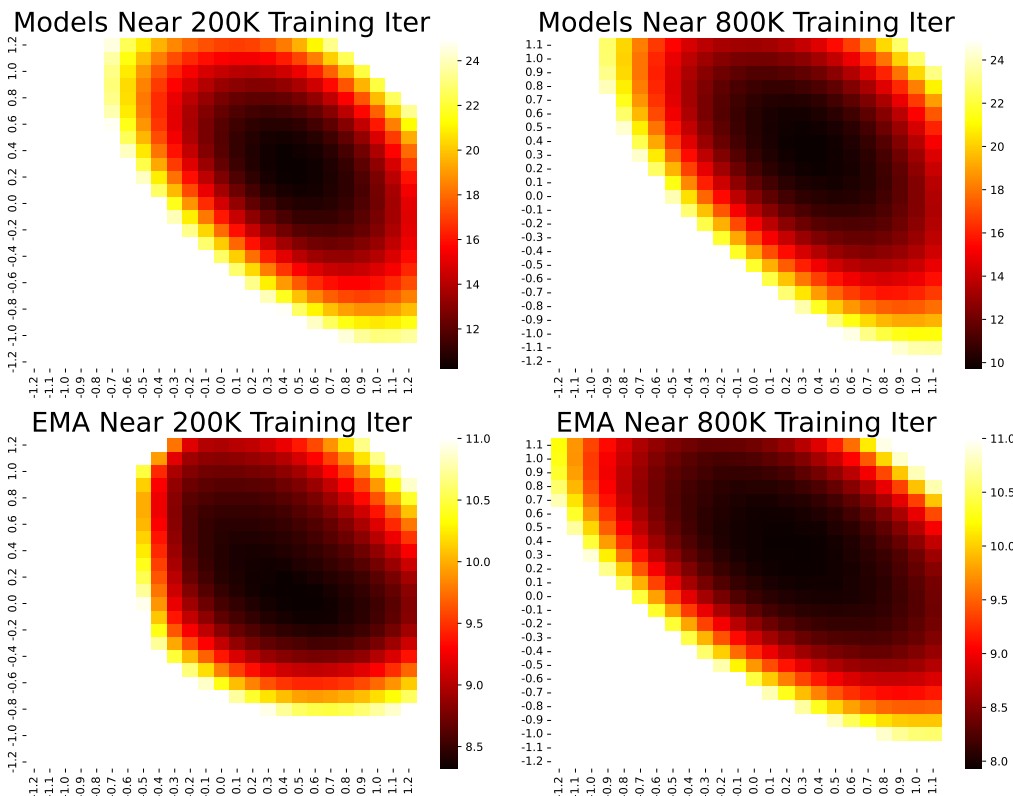

Figure 6: Metric landscapes in grid for CD.

2022) architecture. This decision stems from the official CM code for CIFAR-10 being implemented in JAX while our experiments are conducted using PyTorch. For further details, refer to https://github.com/openai/consistency_models_cifar10. This difference in implementation may partly explain the discrepancies between our replication results and those reported in Table 1 of CM's original paper (Song et al., 2023). On ImageNet-64 and LSUN, we follow the official implementation, employing the ADM (Dhariwal & Nichol, 2021) architecture. Due to resource constraints, we train the model with a smaller batch size of 256 instead of 2048 as used in the original study, resulting in worse outcomes. However, as indicated in Tab. 2, applying LCSC to models trained with a reduced batch size can still achieve performance comparable to models trained with a larger batch size.

### E.1.2 SEARCH SETTING

At each selected training iteration, historical weights are leveraged within designated window sizes and intervals. For CM, checkpoints have a window size of 40K with an interval of 100 on CIFAR-10, and a window size of 20K with an interval of 100 on ImageNet-64 and LSUN datasets. DM is assigned a window size of 50K and an interval of 200 for both datasets. FID calculation for DM on ImageNet-64 and CM on LSUN datasets utilizes a sample of 5K images, while 10K images are used for all other configurations. An evolutionary search spanning 2K iterations is applied consistently across all experimental setups.

### E.1.3 TEXT-TO-IMAGE TASK

For text-to-image task, we fine-tune a LoRA based on the Stable Diffusion v1-5 model (Rombach et al., 2022) on CC12M dataset. We use a batch size of 12 and train the LoRA for 6k steps. For LCSC, we use the checkpoints saved between 4k and 6k steps with an interval of 20. When sampling, we insert LoRA to Dreamshaper-7, which is a fine-tuned version of Stable Diffusion v1-5, and use 1k

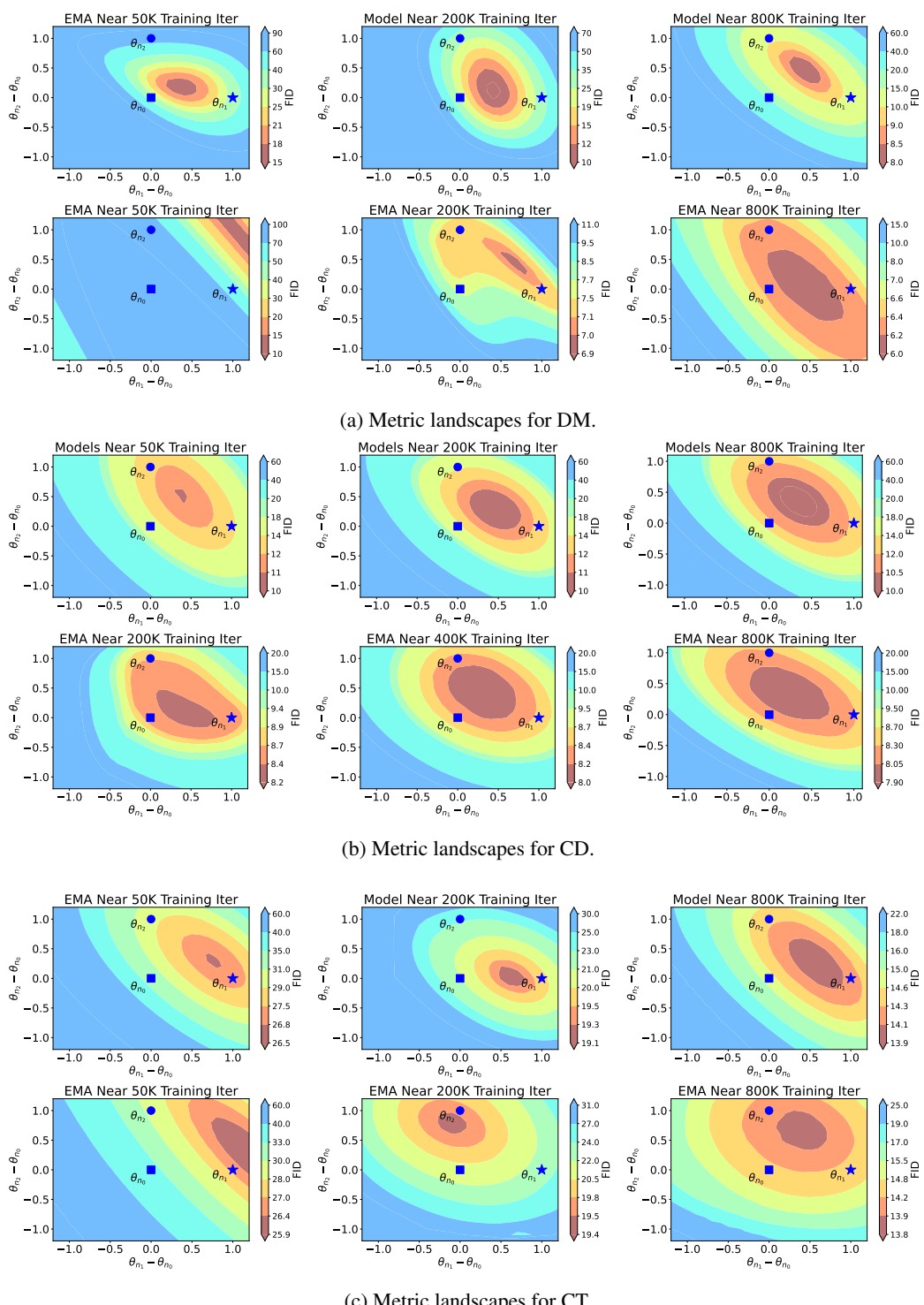

(a) Metric landscapes for DM.

(b) Metric landscapes for CD.

(c) Metric landscapes for CT.

Figure 7: Metric landscapes for DM, CD and CT on CIFAR-10.

samples from CC12M to calculate FID. Finally, we randomly sample 10k samples from MS-COCO dataset to evaluate the search results.

Table 8: Search time consumption per search iteration (in seconds). Results marked with * denote the CPU time cost, which is excludable from the overall time cost through parallel processing. "U-Net" refers to the denoising sampling process, "Inception" refers to the computation of inception features, "Merging" refers to the averaging of weights, and "FID" refers to the computation of FID statistics.

| Model | Dataset | U-Net | Inception | Merging* | FID* |
|-------|---------|-------|-----------|----------|------|
| CM | CIFAR-10 | 7.34 | 4.28 | 5.57 | 2.49 |
|    | ImageNet-64 | 34.7 | 4.32 | 12.0 | 2.37 |
| DM | CIFAR-10 | 40.9 | 4.39 | 4.61 | 2.49 |
|    | ImageNet-64 | 114.5 | 2.24 | 13.8 | 2.53 |

For baselines, we find that the performance is sensitive to the scale of LoRA. Therefore, we first conduct a coarse scan to determine the approximate range of the optimal LoRA scale, followed by a fine sweep. We find 0.15 is the optimal scale for LoRA and use it as the baseline.

## E.2 TWO-PHASE CONVERGENCE

Fig. 8 shows that the generation quality convergence of DM and CM can be divided into two phases. The initial phase is relatively brief, during which DM and CM rapidly acquire the capability to generate visually satisfactory images. In contrast, the second phase is characterized by a slower optimization of models, focusing on the enhancement of sample quality. It is noteworthy that the majority of training iterations belongs to the second phase.

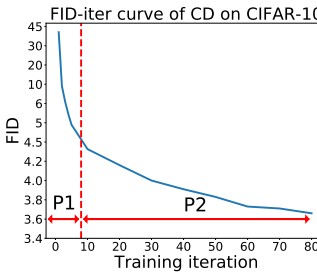 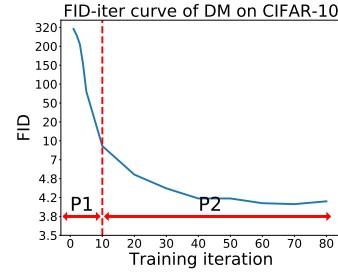

Figure 8: Training curves on CIFAR-10 with CM and DM. P1, P2 represent the first phase and the second phase.

## E.3 SEARCH COST ESTIMATION

We detail the estimation of search costs, covering both CPU and GPU time consumption. Saved checkpoints are loaded into CPU memory and then transferred to the GPU after averaging. Consequently, CPU time comprises the duration for merging weight checkpoints and calculating the FID statistic. Meanwhile, GPU time is dedicated to the sampling and evaluation processes involving the diffusion U-Net and the Inception network. All experiments are performed on a single NVIDIA A100 GPU, paired with an Intel Xeon Platinum 8385P CPU.

First, we profile the durations of each function across all experimental settings and document the time consumed in a single search iteration in Tab. 8. We note that the CPU time cost is significantly lower than that on the GPU. Therefore, while conducting sampling with the current merged weights on the GPU, we can simultaneously perform the FID statistic computation from the previous iteration and merge the checkpoints for the next iteration on the CPU. As a result, the CPU time cost can be effectively excluded from the overall time cost.

Next, we examine the overall training and search time costs as outlined in Tab. 9, and we delve into the speedup ratio detailed in Tabs. 10 and 11. Given that DMs necessitate multiple timesteps for sampling, the overall search cost for DM proves to be non-negligible when compared to the training cost. This results in no observable speedup in convergence upon applying LCSC. However, this search cost is considered manageable and warranted for the anticipated improvements to the final model convergence.

Table 9: Overall search time and training time consumption (in hours).

| Model | Dataset | Search | | Training | | |
|---|---|---|---|---|---|---|
| | | Iteration | Time | Iteration | Batch Size | Time |
| CD | CIFAR-10 | 2K | 6.45 | 800K | 512 | 818 |
| | ImageNet-64 | 2K | 21.7 | 600K | 2048 | 7253 |
| CT | CIFAR-10 | 2K | 6.45 | 800K | 512 | 640 |
| | ImageNet-64 | 2K | 21.7 | 800K | 2048 | 7040 |
| DM | CIFAR-10 | 2K | 25.2 | 800K | 128 | 29.3 |
| | ImageNet-64 | 2K | 64.8 | 500K | 512 | 372 |

Table 10: Accurate training speedup of consistency models on CIFAR-10. The speedup is compared against the standard training with 800K iterations and 512 batch size.

| Model | Method | Training Iter | Batch Size | Speedup($\uparrow$) |
|---|---|---|---|---|
| CD | EMA | 800K | 512 | - |
| | LCSC | 50K | 512 | 14.27$\times$ |
| | | 250K | 512 | 3.12$\times$ |
| | | 100K | 128 | 25.55$\times$ |
| CT | EMA (released) | 800K | 512 | - |
| | LCSC | 400K | 512 | 1.96$\times$ |
| | | 450K | 128 | 6.64$\times$ |

Table 11: Accurate training speedup of consistency models on ImageNet-64. For CD, the speedup is compared against standard training with 600K iterations and 2048 batch size. For CT, the speedup is compared against standard training with 800K iterations and 2048 batch size.

| Model | Method | Training Iter | Batch Size | Speedup($\uparrow$) |
|---|---|---|---|---|
| CD | EMA (released) | 600K | 2048 | - |
| | LCSC | 150K | 256 | 29.20$\times$ |
| | | 300K | 256 | 15.27$\times$ |
| | | 620K | 256 | 7.55$\times$ |
| CT | EMA (released) | 800K | 2048 | - |
| | LCSC | 600K | 256 | 10.33$\times$ |
| | | 1000K | 256 | 6.27$\times$ |

In contrast, for CMs, the search cost is substantially lower than the training cost, which often results in a markedly faster convergence when combining training with LCSC than with training alone. The precise speedup ratios for CM training are detailed in Tabs. 10 and 11. It should be noted that the figures in these tables may exhibit slight discrepancies from those reported in Sec. 5 due to the latter being profiled in an environment with more variables and potential fluctuations.

### E.4 HYPERPARAMETER STUDY

The hyperparameters of LCSC includes the window size ($n_K - n_1$) for retrieving the historical weights, the interval ($n_k - n_{k-1}$) for two adjacent weight checkpoints, the number of samples for computing FID and the number of search iterations, and the number of timesteps for DM.

Tab. 12 illustrates the impact of varying the number of samples, search iterations, and the size of the interval. We observe that increasing the number of samples and search iterations leads to higher performance, though this comes at the expense of increased search cost. In practice, we find that 10K

Table 12: Hyperparameter study of LCSC with (a) DM and (b) CM. The models are trained on CIFAR10 with 250K iterations. We evaluate three values of each hyperparameter and compare them with our adopted setting highlighted in gray. The varied hyperparameter is in bold. For DM the window size is 50K, for CM it is 40K.

|  (a) Results on DM |  |  |  |  |  | (b) Results on CM |  |  |  |
| --- | --- | --- | --- | --- | --- | --- | --- | --- | --- |
| Samples | Search Iters. | Interval | FID($\downarrow$) | IS($\uparrow$) | Samples | Search Iters. | Interval | FID($\downarrow$) | IS($\uparrow$) |
| 10K | 2K | 200 | 3.87 | 9.27 | 10K | 2K | 100 | 2.76 | 9.71 |
| **2K** | 2K | 200 | 9.23 | 9.11 | **2K** | 2K | 100 | 3.40 | 9.39 |
| **5K** | 2K | 200 | 4.04 | 9.09 | **5K** | 2K | 100 | 2.79 | 9.66 |
| 10K | **1K** | 200 | 4.01 | 9.17 | 10K | **1K** | 100 | 3.03 | 9.57 |
| 10K | **4K** | 200 | 3.79 | 9.28 | 10K | **4K** | 100 | 2.69 | 9.70 |
| 10K | 2K | **100** | 3.82 | 9.15 | 10K | 2K | **200** | 2.84 | 9.71 |
| 10K | 2K | **500** | 4.41 | 9.11 | 10K | 2K | **50** | 2.71 | 9.79 |

Table 13: Hyperparameter analysis of LCSC, supplementing the study in Tab. 12. Models are trained on CIFAR10 for 250K iterations. We assess three different values for each hyperparameter, contrasting these with our chosen setting, which is highlighted in gray. The hyperparameter under variation is indicated in bold. For DM, the interval between checkpoints is set to 200, whereas for CM, it is 100. We perform 2K search iterations and sample 10K images to compute the FID score at each iteration.

| Method | Window Size | NFE | FID($\downarrow$) | IS($\uparrow$) |
| --- | --- | --- | --- | --- |
|  | 40K | 1 | 2.76 | 9.71 |
| **CM** | **10K** | 1 | 2.89 | 9.60 |
|  | **50K** | 1 | 2.73 | 9.66 |
|  | 50K | 15 | 3.87 | 9.27 |
|  | **10K** | 15 | 4.41 | 9.11 |
| **DM** | **30K** | 15 | 3.82 | 9.15 |
|  | 50K | **7** | 3.91 | 9.10 |
|  | 50K | **10** | 3.84 | 9.17 |

samples and 2K search iterations can efficiently identify strong models. With a fixed window size, the interval between two checkpoints determines the dimension of the search space. Our findings suggest that search performance generally improves as the search space expands, but limiting the search dimension to fewer than 200 can detrimentally affect search performance.

For the impact of window size $(n_K - n_1)$ and sampling NFE of DM, the findings are detailed in Tab. 13. The results reveal that, despite earlier models being further from convergence, a sufficiently large window for accessing historical weights proves advantageous. Furthermore, conducting a lower NFE during the search in the DM context results in a similar reduction in FID but a smaller improvement in IS. This suggests that the search output's generality across different metrics diminishes when the generated samples during the search are less accurate, i.e., exhibit significant truncation error.

### E.5 DETAILED RESULTS OF EMA RATE GRID SEARCH

For the final training model of each experimental configuration, we conduct a comprehensive sweep across a broad range of EMA rates, presenting the optimal outcomes in Sec. 5. The exhaustive results are detailed in Tab. 14. In the majority of scenarios, the default EMA rate employed in the official implementations of our baseline models (Song et al., 2023; Ho et al., 2020; Nichol & Dhariwal, 2021) yields slightly inferior performance compared to the best EMA rate identified. Nevertheless, the findings demonstrate that LCSC consistently surpasses all explored EMA rates, underscoring the limitations of EMA as a strategy for weight averaging.

Table 14: EMA rate grid search outcomes. Asterisks (*) indicate the results using the default rates from the official paper. Listed FID scores for each EMA rate correspond to fully trained models: CIFAR-10 models at 800K iterations; ImageNet-64 CD/CT/DM models at 600K/1000K/500K iterations, respectively. Iteration counts for models employed by LCSC are shown in parentheses.

| Model | Dataset | EMA rate | | | | | | | LCSC |
|---|---|---|---|---|---|---|---|---|---|
| | | 0.999 | 0.9995 | 0.9999 | 0.999943 | 0.99995 | 0.99997 | 0.99999 | |
| CD | CIFAR-10 | 4.35 | 4.13 | 3.66* | 3.56 | 3.58 | 3.51 | 3.54 | 2.42 (800K) |
| | ImageNet-64 | 7.45 | 7.38 | 7.19 | 7.17* | 7.17 | 7.22 | 7.41 | 5.54 (620K) |
| CT | CIFAR-10 | 9.80 | 9.78 | 9.70* | 9.70 | 9.69 | 9.71 | 9.77 | 8.60 (400K) |
| | ImageNet-64 | 15.7 | 15.7 | 15.6 | 15.6* | 15.6 | 15.6 | 15.7 | 12.1 (600K) |
| DM | CIFAR-10 | 5.73 | 5.18 | 4.16* | 3.99 | 3.96 | 4.04 | 5.04 | 3.18 (800K) |
| | ImageNet-64 | 23.1 | 20.3 | 19.8* | 19.0 | 18.1 | 18.1 | 18.5 | 15.3 (500K) |

Table 15: The comparison of different formulations for base checkpoints on CIFAR-10 dataset. LCSC-Diff stands for the formulation we use in our main experiments. LCSC-Direct stands for using the checkpoints themselves as the weighted base.

| Model | Method | Training Iter | Batch Size | NFE | FID(↓) | IS(↑) | Prec.(↑) | Rec.(↑) | Speed(↑) |
|---|---|---|---|---|---|---|---|---|---|
| CD | LCSC-Direct | 50K | 512 | 1 | 3.18 | 9.60 | 0.67 | 0.58 | ~14× |
| | | 250K | 512 | 1 | 2.76 | 9.71 | 0.67 | 0.59 | ~3.1× |
| | | 800K | 512 | 1 | 2.42 | 9.76 | 0.67 | 0.60 | - |
| | | 800+40K | 512 | 1 | **2.38** | 9.70 | 0.67 | 0.60 | - |
| | | 100K | 128 | 1 | 3.34 | 9.51 | 0.67 | 0.57 | ~23× |
| | LCSC-Diff | 50K | 512 | 1 | 3.10 | 9.50 | 0.66 | 0.58 | ~14× |
| | | 250K | 512 | 1 | 2.66 | 9.64 | 0.67 | 0.59 | ~3.1× |
| | | 800K | 512 | 1 | 2.44 | **9.82** | 0.67 | 0.60 | - |
| | | 800+40K | 512 | 1 | 2.50 | 9.70 | 0.68 | 0.59 | - |
| | | 100K | 128 | 1 | 3.21 | 9.48 | 0.66 | 0.58 | ~23× |
| CT | LCSC-Direct | 400K | 512 | 1 | 8.60 | 8.89 | 0.67 | 0.47 | ~1.9× |
| | | 800+40K | 512 | 1 | 8.05 | 8.98 | 0.70 | 0.45 | - |
| | | 450K | 128 | 1 | 8.33 | 8.67 | 0.69 | 0.44 | ~7× |
| | LCSC-Diff | 400K | 512 | 1 | 8.89 | 8.79 | 0.67 | 0.47 | ~1.9× |
| | | 800+40K | 512 | 1 | **7.05** | **9.01** | 0.70 | 0.45 | - |
| | | 450K | 128 | 1 | 8.54 | 8.66 | 0.69 | 0.44 | ~7× |

## E.6 DIFFERENT FORMULATION OF THE BASE CHECKPOINTS

As discussed in Sec. 4.2, we subtract the first checkpoint from each checkpoint and use these differences as the weighted base. In this section, we explore an alternative approach for defining the base checkpoints: using all the checkpoints themselves as the base. In this case, the final combined weight is:

$$\boldsymbol{\alpha} \cdot \Theta = \sum_{i=1}^{K} \alpha_i \boldsymbol{\theta}_{n_i}. \tag{77}$$

We apply this formulation in our search, with the results presented in Tab. 15 and Tab. 16. We can see that the difference search method outperforms the checkpoint search method in all cases on the ImageNet-64. On CIFAR-10, the performance of the two methods are similar with each other.

## E.7 EVALUATION WITH OTHER METRICS

## E.7.1 EVALUATION WITH FCD AND KID

In this section, we present the results of FCD and KID using CT method on CIFAR-10 and ImageNet-64 dataset. The results are shown in Tab. 17.

Table 16: The comparison of different formulations for base checkpoints on ImageNet-64 dataset. LCSC-Diff stands for the formulation we use in our main experiments. LCSC-Direct stands for using the checkpoints themselves as the weighted base.

| Model | Method | Training Iter | Batch Size | NFE | FID(↓) | IS(↑) | Prec.(↑) | Rec.(↑) | Speed(↑) |
|---|---|---|---|---|---|---|---|---|---|
| CD | LCSC-Direct | 300K | 256 | 1 | 5.71 | 41.8 | 0.68 | 0.62 | ∼15× |
| | | 600+20K | 256 | 1 | 5.54 | 40.9 | 0.68 | 0.62 | ∼7.6× |
| | LCSC-Diff | 300K | 256 | 1 | 5.51 | 39.8 | 0.68 | 0.62 | ∼15× |
| | | 600+20K | 256 | 1 | **5.07** | **42.5** | 0.69 | 0.62 | ∼7.6× |
| CT | LCSC-Direct | 600K | 256 | 1 | 12.1 | 35.1 | 0.67 | 0.54 | ∼10.4× |
| | | 800K | 256 | 1 | 11.1 | 35.7 | 0.65 | 0.57 | ∼6.3× |
| | LCSC-Diff | 600K | 256 | 1 | 10.5 | 36.8 | 0.66 | 0.56 | ∼10.4× |
| | | 800K | 256 | 1 | **9.02** | **38.8** | 0.68 | 0.55 | ∼7.3× |

Table 17: Evaluation results with FCD and KID metrics on CIFAR-10 and ImageNet-64 datasets.

(a) Results on CIFAR-10.

| Method | Training Iter | Batch Size | FID(↓) | FCD(↓) | KID(↓) |
|---|---|---|---|---|---|
| CT | 400k | 512 | 12.1 | 43.0 | 7.47e-3 |
| CT | 800k | 512 | 9.87 | 35.8 | 5.14e-3 |
| LCSC | 450k | 128 | 8.54 | 27.2 | 4.65e-3 |
| LCSC | 400k | 512 | 8.89 | 34.4 | 3.96e-3 |
| LCSC | 800+40k | 512 | **7.05** | **24.9** | **2.90e-3** |

(b) Results on ImageNet-64.

| Method | Training Iter | Batch Size | FID(↓) | FCD(↓) | KID(↓) |
|---|---|---|---|---|---|
| CT | 600k | 256 | 16.6 | 49.8 | 9.22e-3 |
| CT | 800k | 256 | 15.8 | 47.5 | 8.69e-3 |
| CT | 800k | 2048 | 13.1 | 47.5 | 8.55e-3 |
| LCSC | 600k | 256 | 10.5 | 38.8 | 4.41e-3 |
| LCSC | 800k | 256 | **9.02** | **30.0** | **3.87e-3** |

### E.7.2 EVALUATION WITH FID ON TEST DATASET

In this section, we report the FID calculated on the test dataset to validate that LCSC does not over-fit on the training data. Results are shown in Tab. 18. We can see that LCSC also achieves significant improvement on test FID, indicating its generalization ability across different data.

### E.8 SEARCH RESULTS WITH STABLE DIFFUSION MODELS

To further demonstrate the effectiveness of LCSC in enhancing DMs, we conduct experiments on Stable Diffusion (Rombach et al., 2022) checkpoints. Specifically, we fine-tune the Stable Diffusion v1-5 model on CC12M using LoRA for 20k iterations. We then apply LCSC to the saved check-points at intervals of 100 iterations. The results, presented in Tab. 19, show that LCSC achieves a significant improvement compared to the released Stable Diffusion checkpoints with the same sampling NFE.

To investigate whether LCSC can accelerate the Stable Diffusion model, we further test PickScore (Kirstain et al., 2023) between the LCSC model with 10 NFE and the Stable Diffusion model with 15 NFE. The results are 0.49 and 0.51 for Stable Diffusion and LCSC, respectively, with a 57% winning rate for LCSC. This further demonstrate the ability of LCSC to accelerate inference speed for diffusion models.

## F MORE INSIGHTS AND ANALYSIS

### F.1 ANALYSIS OF SEARCH PATTERNS

#### F.1.1 MORE EXAMPLES OF SEARCH PATTERNS

We demonstrate search results on ImageNet-64 in Fig. 9, which share similar patterns with results on CIFAR-10 (refer to Fig. 5). Additionally, Fig. 10 shows that the checkpoints assigned with larger coefficients often has lower FID than the checkpoints with small coefficients.

#### F.1.2 SEARCH PATTERN OF DIFFERENT RANDOM SEED

We compare search patterns across different random seeds. As shown in Fig. 11, although each seed's dominant coefficients map to distinct subsets of weight checkpoints, they produce similar outcomes in terms of FID as provided in the caption. These results suggest multiple high-quality

Table 18: Evaluation results on CIFAR-10 test set.

| Method | Training Iter | Batch Size | FID(↓) | test FID(↓) |
|--------|---------------|------------|--------|-------------|
| CD | 800k | 512 | 3.66 | 5.88 |
| CD | 840k | 512 | 3.65 | 5.85 |
| LCSC | 100k | 128 | 3.21 | 5.48 |
| LCSC | 250k | 512 | 2.66 | 4.88 |
| LCSC | 800k | 512 | **2.44** | **4.70** |
| LCSC | 800+40k | 512 | 2.50 | 4.75 |

Table 19: Results of LCSC on Stable Diffusion Checkpoints.

| | NFE | FID | PKS | WR@PKS | CLIP Score |
|---|-----|-----|-----|--------|------------|
| LCSC | 15 | **16.30** | **0.53**(v.s. SD)/**0.53**(v.s. LoRA) | **55%**(v.s. SD)/**56%**(v.s. LoRA) | **26.69** |
| SDv1-5 | 15 | 17.55 | 0.47 | 44% | 26.60 |
| LoRA tuning | 15 | 17.05 | 0.47 | 45% | 26.61 |
| LCSC | 10 | **16.68** | **0.59**(v.s. SD)/**0.51**(v.s. LoRA) | **64%**(v.s. SD)/**53%**(v.s. LoRA) | **26.61** |
| SDv1-5 | 10 | 18.16 | 0.41 | 36% | 26.57 |
| LoRA tuning | 10 | 17.35 | 0.49 | 47% | 26.56 |

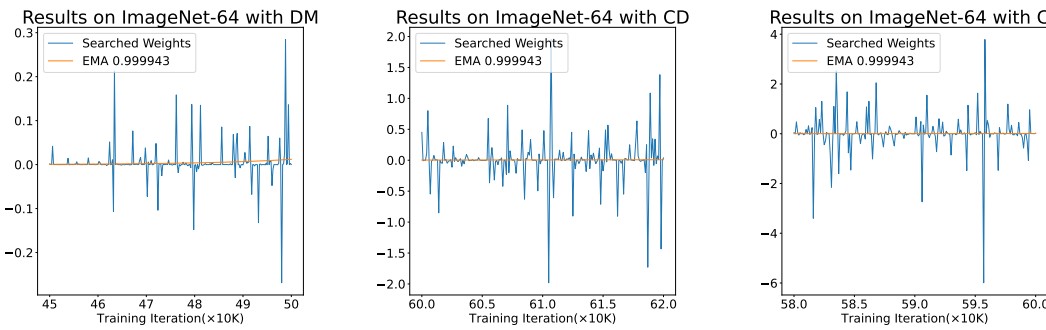

Figure 9: Visualization of weight combination coefficients obtained using LCSC compared to those from the default EMA on ImageNet-64.

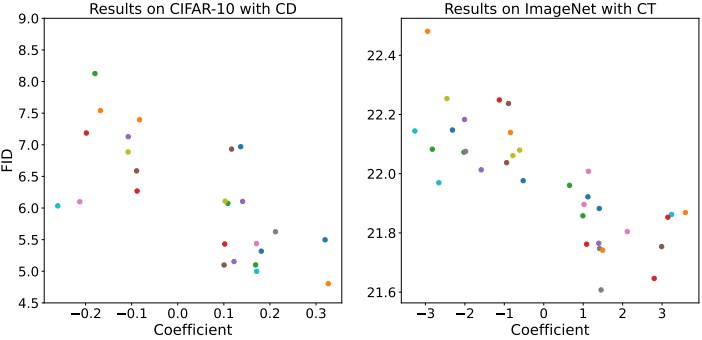

Figure 10: FID of checkpoints with varying magnitude of coefficients.

basins exist within the weight checkpoint subspace, and LCSC converges to one of them depending on the randomness. Previous study demonstrates that local minimal in networks are connected by simple curves over which losses are nearly constant (Garipov et al., 2018). To investigate whether solutions found by LCSC are also connected we average the search patterns obtained from multiple random seeds and evaluate the performance of the resulting model. As shown in Fig. 12, the averaged search pattern becomes homogeneous and contains smaller coefficients, yet it achieves comparable performance, as detailed in the figure caption. This observation suggests that the search results of LCSC may also be connected by low-loss curve.

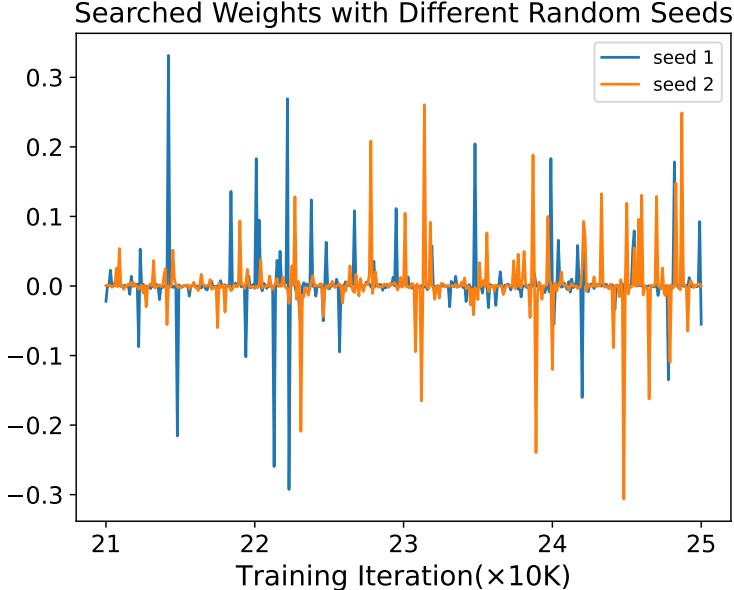

Figure 11: Visualization of linear combination coefficients obtained by LCSC with different random seeds using CD on CIFAR10. Their FID scores are: 2.76 (seed 1) and 2.69 (seed 2).

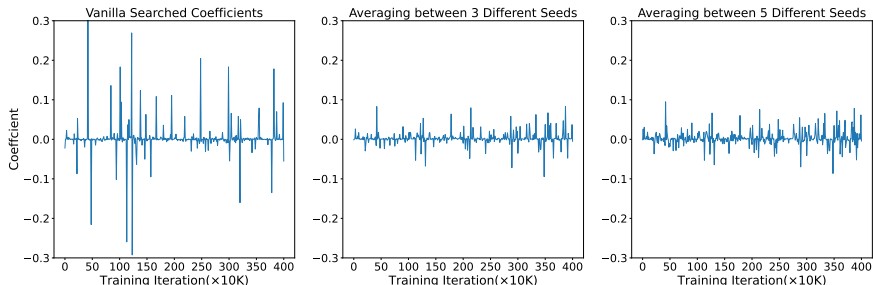

Figure 12: Visualization of averaged coefficients obtained by LCSC across different random seeds using CD on CIFAR10. Their FID scores are: 2.76 (left), 2.70 (middle), 2.83 (right).

### F.1.3 THE CRITICAL ROLE OF SEARCH SPACE CONFIGURATION

As explored in Sec. 6, restricting averaged weights to the convex hull of candidate checkpoints, where all combination coefficients are non-negative and their sum equals 1, might limit search efficacy. To test this hypothesis, we perform searches under convex combination conditions by clipping coefficients to non-negative values and normalizing their sum to 1. Tab. 20 presents these findings, with "w/o" indicating no restriction on coefficients (allowing values below zero) and "w" representing the convex hull restriction. The results clearly show superior performance without the convex restriction, underscoring its limiting effect on search outcomes. Furthermore, we discover that normalizing the coefficient sum to 1 is vital for effective search exploration, a practice we continue even after lifting the convex condition, as detailed in Eq. (7).

### F.1.4 FORMATION OF THE SEARCH PATTERN

We observe that the search pattern is established during the early stage of the search, while the later stage primarily amplifies it to a certain magnitude, as shown in Fig. 13.

Table 20: Performance of LCSC with or without restriction to the convex hull of all saved checkpoints. "w/o" in the column "Restriction" is our default setting in main experiments, while "w" means the restriction holds.

| Model | Method | Training Iter | Batch Size | NFE | Restriction | FID($\downarrow$) | IS($\uparrow$) |
|---|---|---|---|---|---|---|---|
| **CD** | **LCSC** | 250K | 512 | 1 | w/o | 2.76 | 9.71 |
| | | | | | w | 3.38 | 9.36 |
| **DM** | **LCSC** | 350K | 128 | 15 | w/o | 3.56 | 9.35 |
| | | | | | w | 3.59 | 9.29 |

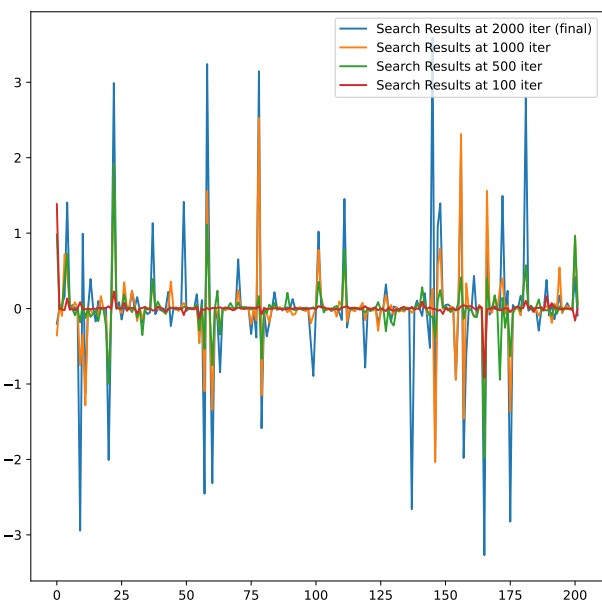

Figure 13: Visualization of coefficient pattern at different number of search iterations. The experiment is conducted on ImageNet using CT.

## F.2 REGULARIZED EVOLUTIONARY SEARCH

We conduct additional experiments to evaluate the impact of regularization. Specifically, during the search process, we clip all coefficients to be below 1, thereby constraining the search to a restricted space. Using the same initialization and random seed, Fig. 14 shows that LCSC produces smaller and more homogeneous coefficients with regularization compared to those obtained without it. However, as noted in the figure caption, applying regularization results in worse performance, suggesting that constraining LCSC to smaller and more homogeneous coefficients is not advantageous.

## F.3 CONVERGENCE CURVE OF LCSC

Fig. 15a illustrates the convergence curve of the models searched by LCSC. At each point, corresponding to a specific number of training iterations, we perform LCSC with 2K search iterations using the most recent checkpoints available at that point. The results indicate that as the number of training iterations increases, the models searched by LCSC converge to progressively lower FID values. Notably, the convergence curve of LCSC exhibits a similar trend to that of the EMA model's

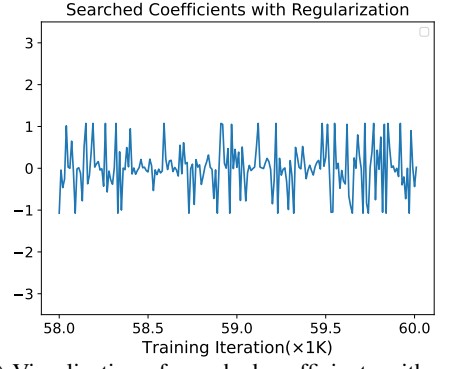 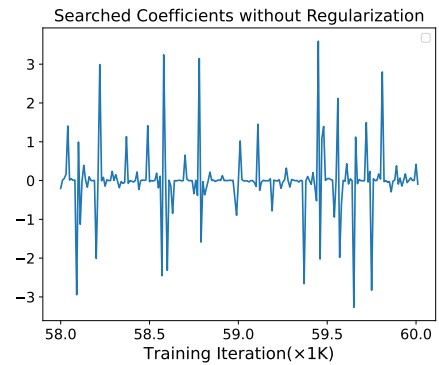

(a) Visualization of searched coefficients with regularization.

(b) Visualization of searched coefficients without regularization.

Figure 14: Visualization of searched coefficients with and without regularization. Respective FID/IS/Prec/Rec are: 13.40/33.4/0.64/0.55 (left), 10.5/36.8/0.66/0.56 (right).

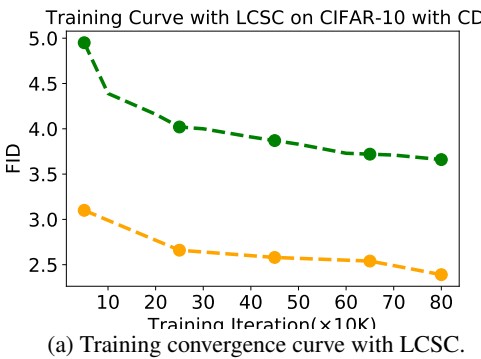 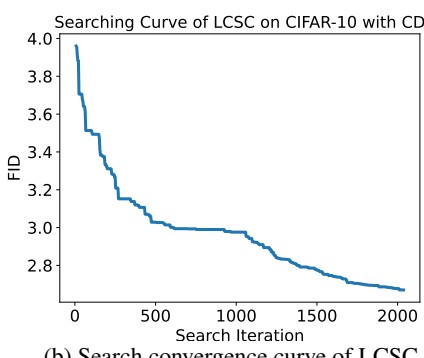

(a) Training convergence curve with LCSC.

(b) Search convergence curve of LCSC.

Figure 15: Convergence curves of LCSC on CIFAR-10 with CD.

convergence curve. However, LCSC consistently achieves lower FID values compared to the EMA model, highlighting its enhanced ability to effectively merge historical checkpoints. Additionally, Fig. 15b shows the convergence curve of LCSC iteself. This curve consistently reduces as the number of search iteration increases.

### F.4  DISCUSSION ON THE TRAINING VARIANCE OF DM AND CM

In Sec. 3, we mentioned that the training objectives of DM and CM tend to introduce substantial variance in gradient estimations. We further discuss the potential reasons or hypothesis of such phenomenon as follows.

#### F.4.1  UNBIASED ESTIMATION AS OBJECTIVE

The objectives used in DM and CT are not accurate for any single batch. Instead, they serve as an unbiased estimation. Specifically, for DM, the neural network learns the score function at any $(\boldsymbol{x}_t, t)$, denoted as $\boldsymbol{s}_\theta(\boldsymbol{x}_t, t)$. The ground truth score function $\nabla \log p_t(\boldsymbol{x}_t)$ is given as (Song et al., 2020b):

$$\nabla \log p_t(\boldsymbol{x}_t) = \mathbb{E}_{\boldsymbol{y} \sim p_{\text{data}}, \varepsilon \sim \mathcal{N}(\mathbf{0}, \sigma(t)^2 \mathbf{I})} \left( -\frac{\varepsilon}{\sigma_t} \big| \alpha_t \boldsymbol{y} + \sigma_t \varepsilon = \boldsymbol{x}_t \right) \tag{78}$$

This formula implies that every sample $\boldsymbol{y}$ in the dataset contributes to the ground truth score function at any $(x_t, t)$. However, the training objective Eq. (2) provides only an unbiased estimation of the ground truth score Eq. (78). At every iteration, the current target for the network is not identical to the ground truth score function but is determined by the sample $\boldsymbol{y}$ randomly drawn from the dataset and the noise level $\sigma_t$. Thus, even if $\boldsymbol{s}_\theta(\boldsymbol{x}_t, t)$ exactly matches $\nabla p_t(\boldsymbol{x}_t)$ for $\forall(\boldsymbol{x}_t, t)$, its gradient estimation using a mini-batch of data does not converge to zero. The approximation of ground truth score function can only be obtained through the expectation over many training iterations.

For CT, the situation is very similar. To simulate the current ODE solution step without a teacher diffusion model, Song et al. (Song et al., 2023) utilize the Monte Carol estimation $-\frac{\boldsymbol{x}_t - \boldsymbol{y}}{t^2}$ to replace the ground truth score function $\nabla \log p_t(\boldsymbol{x}_t) = \mathbb{E}_{\boldsymbol{y} \sim p_{\text{data}}, \boldsymbol{x}_t \sim \mathcal{N}(\boldsymbol{y}, t^2 \mathbf{I})}(-\frac{\boldsymbol{x}_t - \boldsymbol{y}}{t^2} | \boldsymbol{x}_t)$ under the noise schedule from EDM (Karras et al., 2022) as denoted in Eq. (4). Therefore, the objective is also not accurate for any single batch of data and the model has to learn to fit the target model output after solving the current step with the expectation of score function among many training iterations. These types of objectives, which are not accurate for any single batch, introduce high variance to the gradient estimation.

### F.4.2 MODEL ACROSS DIFFERENT TIME STEPS

For both DM and CM, the neural network has to be trained at various time steps. Since the model input and output follow different distributions, the gradients at different time steps may conflict with each other. This phenomenon of negative transfer between timesteps has been studied in several previous work, where they train different diffusion neural networks at different timesteps and achieve better performance (Hang et al., 2023; Go et al., 2024). The misalignment between the objectives of different time steps may also contribute to the high variance in the gradient estimation.

### F.4.3 ERROR ACCUMULATION OF CM

For CM, the model learns to approximate the output of the target model at the previous step. This could potentially introduce the problem of error accumulation (Berthelot et al., 2023). Consequently, any noise introduced during training at early time steps is likely to lead to inaccuracies in the target model, which may be magnified in subsequent timesteps. This property of CM amplifies the high training variance, particularly for one-step sampling.

## G FUTURE WORK

LCSC represents a novel optimization paradigm, indicating its potential for widespread application. We recommend future investigations focus on three key areas:

- *Expanded Search Space:* Presently, LCSC applies a uniform coefficient across an entire model. Considering that different model layers might benefit from distinct combination coefficients, partitioning model weights into segments for unique coefficient assignments could enhance the search space. For DMs, adopting variable coefficients across different timesteps may also offer further improvements.
- *Efficient Optimization Methods:* The current reliance on evolutionary methods, characterized by their dependency on randomness, limits efficiency and risks convergence to local optima. Investigating more effective optimization strategies presents a promising avenue for enhancing LCSC.
- *Broader Application Scope:* While the initial motivation for LCSC stems from managing the high training variance observed in DMs and CMs, its utility is not confined to these models. Exploring LCSC's applicability to other domains, such as fine-tuning language models or additional vision models, could unlock new performance gains.

## H VISUALIZATION

In this section, visualizations of images generated by LCM-LoRA (Fig. 16), DM (Figs. 17 and 18), CD (Figs. 19 and 20), and CT (Figs. 21 and 22) are presented. Many images produced by the EMA model are observed to be similar to those produced by the model derived from our LCSC when using the same noise input. This is expected given that both models are based on weights from the same training cycle. To more clearly highlight the distinctions between EMA and LCSC, we generate 50K images from EMA and LCSC using the same set of noise inputs, order the image pairs according to their Euclidean distances in the inception feature space, and randomly select the images with large distances. In general, images generated by the LCSC model are found to display enhanced sharpness, diminished noise, and more distinct object representation as well as details.

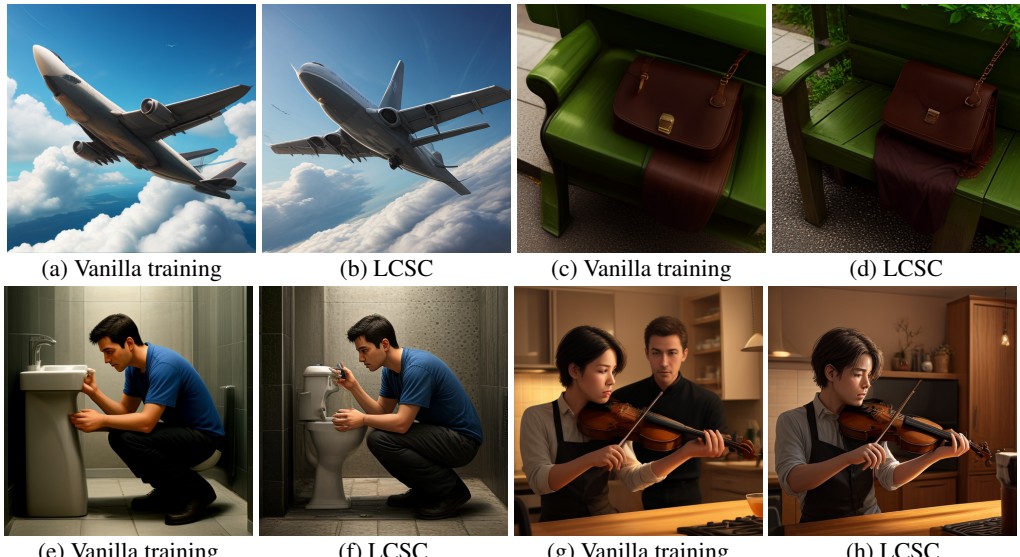

Figure 16: Examples of the images generated by LCSC and vanilla training. The prompts for the images are 'A large passenger airplane flying through the air', 'A brown purse is sitting on a green bench', 'A man getting a drink from a water fountain that is a toilet', 'A picture of a man playing a violin in a kitchen' respectively.

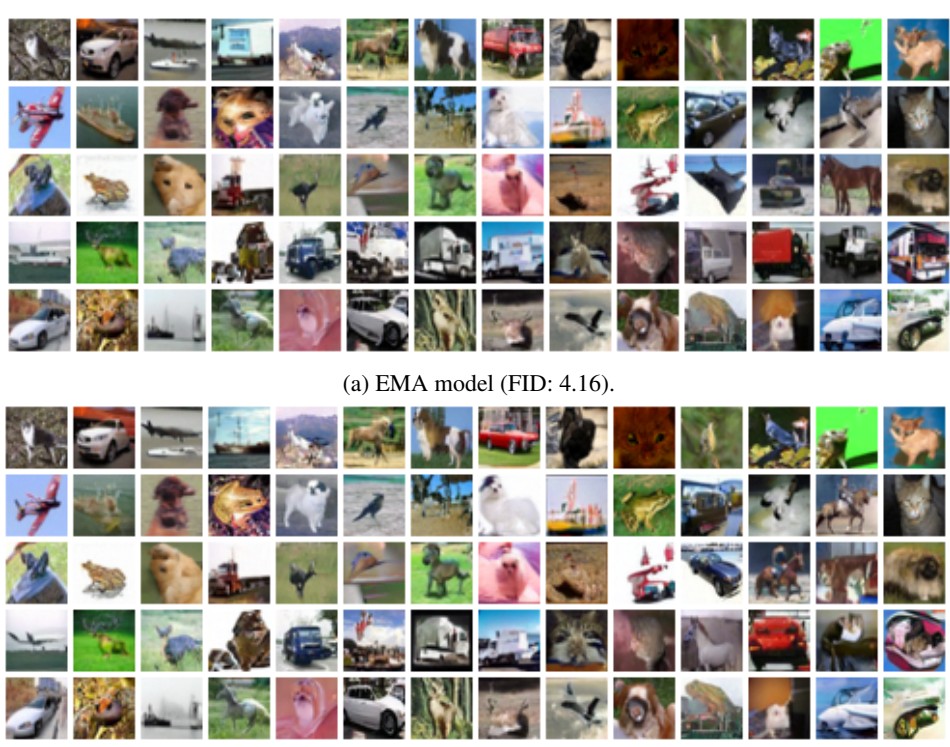

(b) Model obtained through LCSC (FID: 3.18).

Figure 17: Generated images from a DM trained on CIFAR10 over 800K iterations.

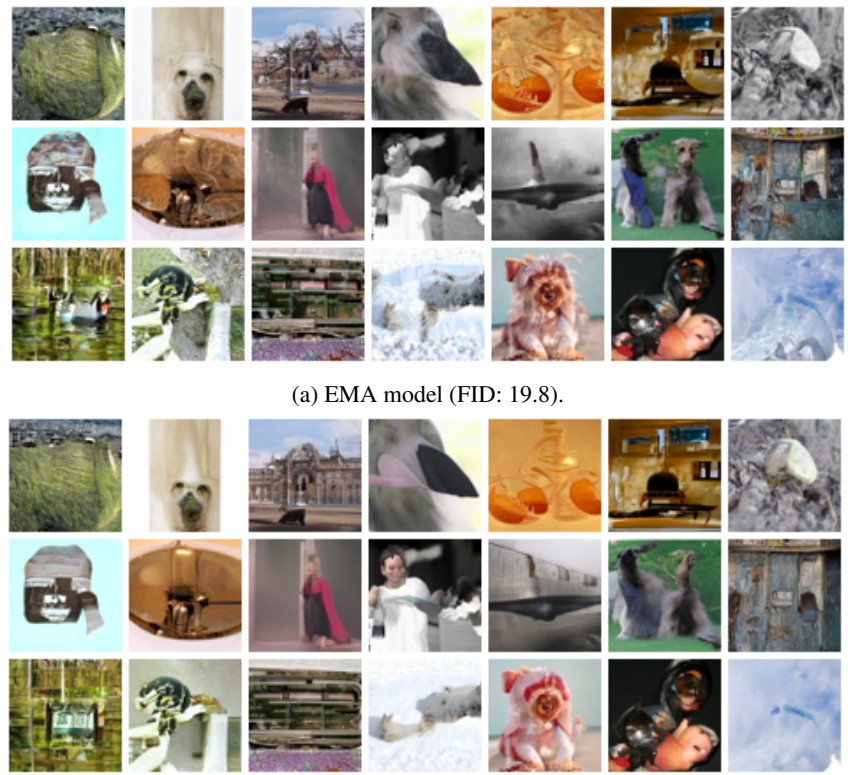

(a) EMA model (FID: 19.8).

(b) Model obtained through LCSC (FID: 15.3).

Figure 18: Generated images from a DM trained on ImageNet-64 over 500K iterations.

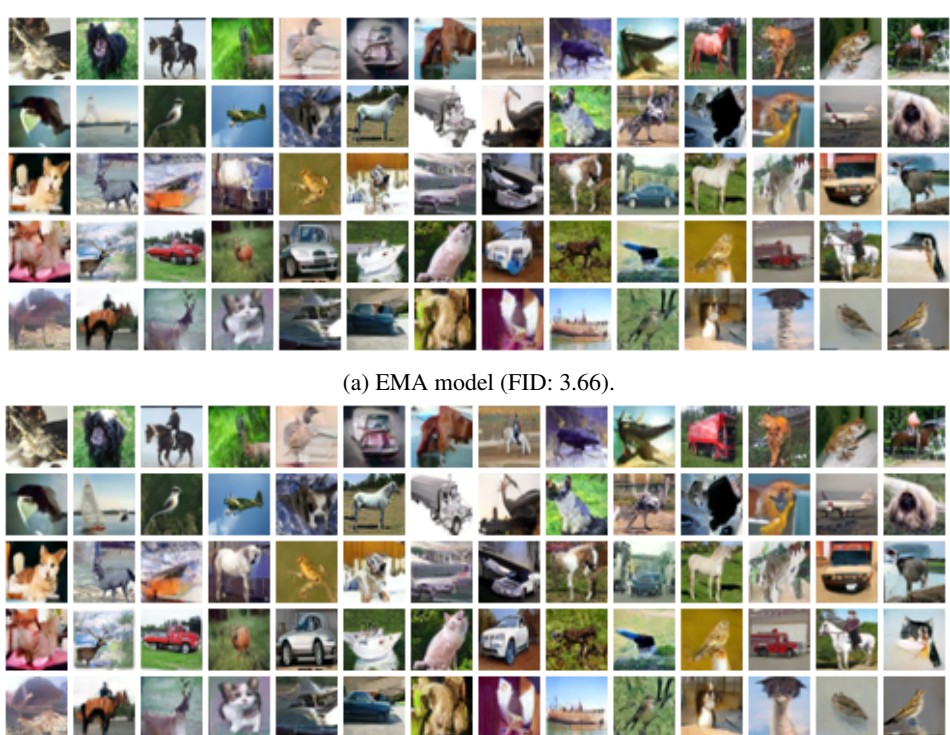

(a) EMA model (FID: 3.66).

(b) Model obtained through LCSC (FID: 2.42).

Figure 19: Generated images from a CD model trained on CIFAR10 for 800K iterations.

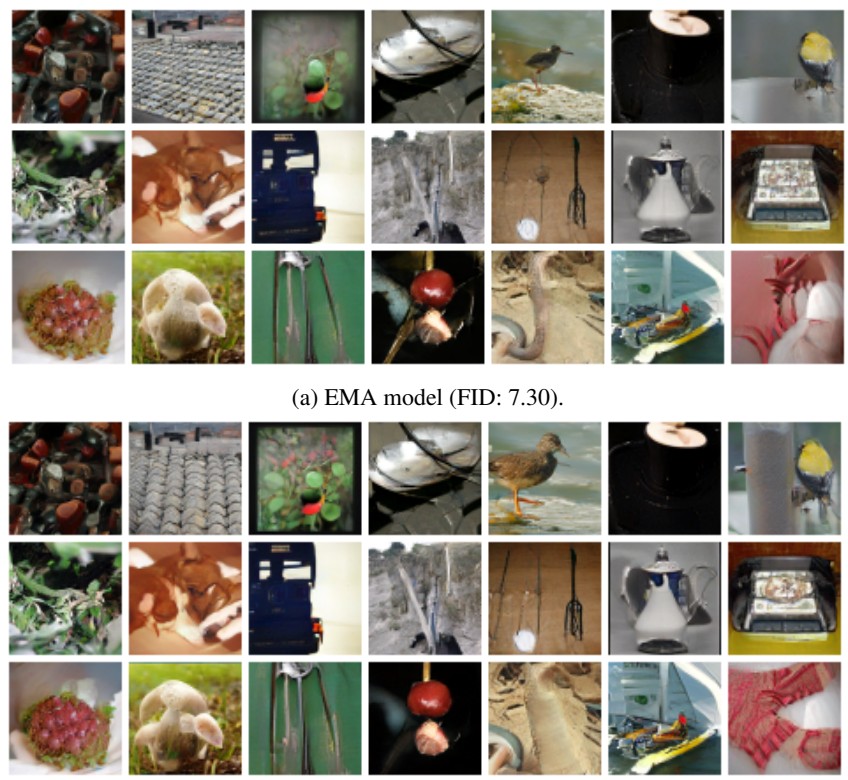

(a) EMA model (FID: 7.30).

(b) Model obtained through LCSC (FID: 5.54).
Figure 20: Images from a CD model trained on ImageNet-64 for 600K iterations.

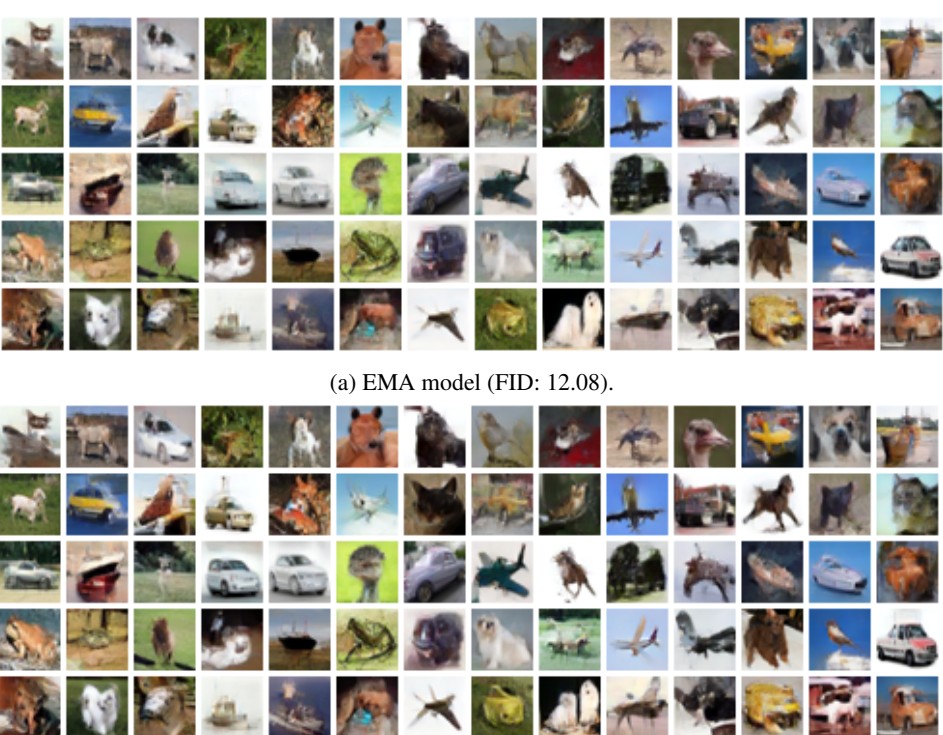

(a) EMA model (FID: 12.08).

(b) Model obtained through LCSC (FID: 8.60).
Figure 21: Generated images from a CT model trained on CIFAR10 for 400K iterations.

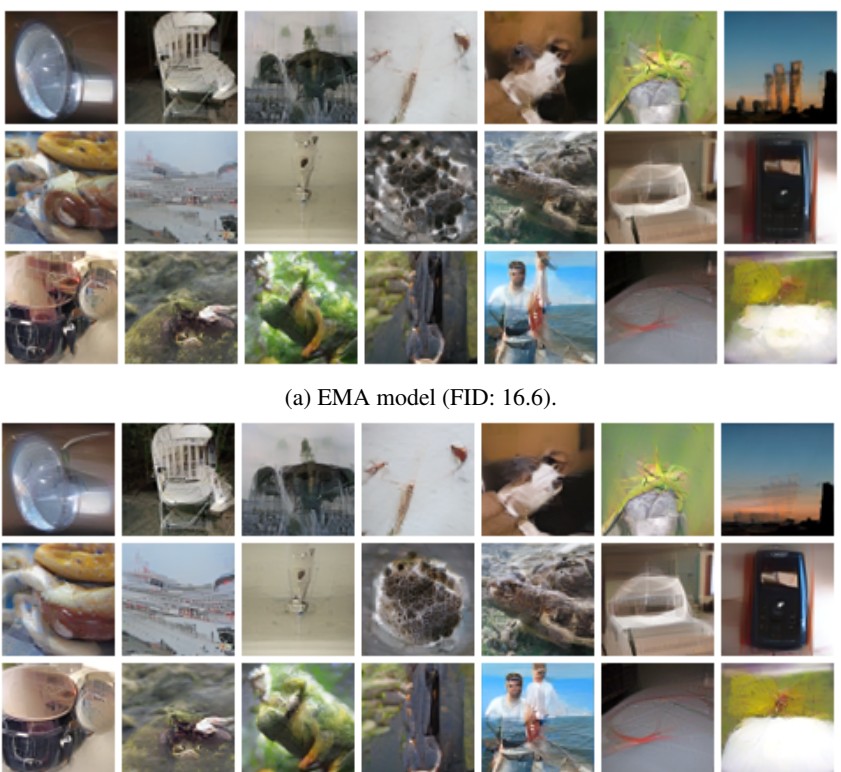

(a) EMA model (FID: 16.6).

(b) Model obtained through LCSC (FID: 12.1).

Figure 22: Images from a CT model trained on ImageNet-64 for 600K iterations.

