# OpenReview forum: "Linear Combination of Saved Checkpoints Makes Consistency and Diffusion Models Better"
_ICLR.cc/2025/Conference — ICLR 2025 Poster_

### Official Review · Reviewer_AvHi · 2024-10-30

**Soundness:** 3
**Presentation:** 4
**Contribution:** 3
**Rating:** 6
**Confidence:** 3

**Summary:**

This paper targets the efficient training problem for diffusion models and consistency models. They find that proper checkpoint merging can significantly improve the training convergence and final performance. Therefore they develop an evolutionary search algorithm for linearly combining the checkpoints. The experiments showcase their LCSC can reduce the training cost and enhance the performance of pre-trained diffusion models.

**Strengths:**

1. The finding that proper checkpoint merging can significantly improve the training convergence and final performance is novel and interesting to me.
2. The theoretical analyses for EMA are promising and solid.
3. The experiments are extensive and convincing.

**Weaknesses:**

1. The details for how to get the metric landscapes are missing. Do you apply a grid search for all $x,y$ with the formula $\theta_{(x, y)}=\theta_{n_0}+x\left(\theta_{n_1}-\theta_{n_0}\right)+y\left(\theta_{n_2}-\theta_{n_0}\right)$. If so, what is the grid interval? And what is the interval between $n_0$, $n_1$ and $n_2$? It is quite strange that the optimal points are always located inside of the regions surrounded by $n_0$, $n_1$, and $n_2$. Does it mean the models are always oscillating? Please provide more explanation.
2. The claim in lines 528-529 that "a small subset of weights is characterized by coefficients of large magnitude" is interesting but needs more evidence.
3. Please provide more insights about why the coefficients of some iterations in Figure 5 are negative. It is clear that the negative coefficients mean the models at these iterations have a negative effect on the final performance but why does this happen? Is it because the models at these iterations have a relatively lower FID?

**Questions:**

1. How to get such smooth metric landscapes?
2. How to prove "a small subset of weights is characterized by coefficients of large magnitude"?
3. Why do the models at some iterations have negative coefficients?

---

> ### Author Response · Authors · 2024-11-25
> **Response to Reviewer AvHi 1**
>
> We sincerely thank the reviewer for their thoughtful feedback and for recognizing the novelty of our findings, the solidity of our theoretical analysis of EMA, and the extensive and convincing experimental validations of LCSC. We appreciate the acknowledgment of our method's effectiveness in improving training convergence, final performance, and pre-trained model enhancement.
>
> We have revised our paper with all updated contents in **orange**. Here are our responses to your concerns and questions:
>
> **W1 & Q1:** The details for how to get the (smooth) metric landscape.
>
> - **Experimental Settings**: We indeed apply a grid search to get the performance of $\theta(x,y)$. We set the grid size as **0.1**. We choose 5k as the interval between $n_0$, $n_1$ and $n_2$, i.e. $n_1 = n_0 + 5000$, $n_2 = n_1 + 5000$.
> - **Explanation of the Phenomenon**: The training objective of DM and CM tends to introduce high variance in gradient estimation, as we discuss in Appendix F.2. Many factors like the unbiased estimation form of loss (for DM and CT), the use of target model (for CD and CT) and the large number of timesteps (for DM, CD and CT) may lead to the high variance. So it is reasonable for the model to oscillate during training. Additionally, the widely validated fact that EMA is more effective than the model itself in training DM or CM also supports this point.
>
> - After obtaining the grid search results, we utilize the 'contourf` function in the matplotlib.pyplot package, which will produce smooth landscapes for better visualization. To address your concern, we further demonstrate heatmaps strictly arranged in a grid pattern in Figure 6.
>
> **W2 & Q2:** "A small subset of weights is characterized by coefficients of large magnitude" is interesting but needs more evidence. How to prove?
>
> -  This claim is supported by our experimental results, and we have provided multiple pieces of empirical evidence, including visualizations of the sparse and multi-peak coefficient patterns in Figures 5, 9, 11, and 12. If the reviewer would like to see a specific form of evidence, we are open to conducting additional experiments and presenting the results.
>
> -  Notably, we observe that better performance may still be achievable with smaller and more homogeneous coefficients, although the evolutionary search may not be well-suited for regularization in this application scenario. To demonstrate that, we average the searched coefficients across multiple random seeds. As shown in Figure 12 in the revision, the averaged coefficients become smaller and more homogeneous, while the resulting model performs similarly well as every single searched model as indicated in the figure caption.
>
> -  The above observation raises an interesting question: why does LCSC prefer solutions with large coefficients rather than the ones with smaller and more homogeneous coefficients? We hypothesize that this phenomenon might be due to our implementation choice of the evolutionary search algorithm. Specifically, during the mutation phase, the standard deviation of the Gaussian noise we added to each *coefficient* is proportional to its current scale. Therefore, large coefficients tend to move faster, and small coefficients tend to stay stably small. This would make "smaller, more homogeneous coefficients" less likely to happen than coefficients with sparse patterns.
>
> **W3 & Q3:** Why do the models at some iterations have negative coefficients? Is it because the models at these iteration have a relatively lower FID
> -  Thank you for your interest in gaining deeper insights into the searched coefficients. To provide further analysis, we evaluated the FID of checkpoints with large positive and large negative coefficients. Interestingly, checkpoints with large negative coefficients tend to exhibit worse FID scores, as shown in Figure 10 of the revision.
>
> -  From an algorithmic perspective, because we constrain the sum of coefficients to one—a design choice aimed at improving search efficiency—large positive coefficients inherently require equally large negative coefficients to balance the sum. As previously discussed, our evolutionary search naturally favors sparse solutions. As a result, we observe a few large negative coefficients compensating for the large positive coefficients.

---

> > ### Comment · Reviewer_AvHi · 2024-11-26
> >
> > Thank you very much for your detailed rebuttal. My concerns have been addressed. I will maintain my original score which is positive toward this paper.

---

### Official Review · Reviewer_Enc2 · 2024-11-04

**Soundness:** 3
**Presentation:** 3
**Contribution:** 3
**Rating:** 6
**Confidence:** 3

**Summary:**

The authors introduce a novel approach to enhance the performance and reduce the training costs of Diffusion Models (DM) and Consistency Models (CM) by learning a linear combination of saved checkpoints using evolutionary algorithms. The paper builds upon existing theoretical frameworks, demonstrating that Exponential Moving Average (EMA) models converge faster than their last-tier counterparts under specific conditions. Through a motivational experiment involving a grid search over numerous linear combinations of three checkpoints (with and without EMA), the authors reveal the existence of superior solutions compared to the traditional EMA approach. The proposed method, termed Linear Combination of Saved Checkpoints (LCSC), is further validated experimentally, showing significant reductions in training costs via fewer iterations and smaller batch sizes, as well as improvements in generation quality when applied to fine-tuned checkpoints.

**Strengths:**

1. Impressive Experimental Results: The experiments, particularly those presented in Tables 1 and 2, convincingly demonstrate the efficacy of LCSC for both CD and CT, showcasing significant reduction in training costs.
2. Innovative Findings: Figures 4 and 6 highlight the potential for significantly better linear combinations of checkpoints than the naive EMA solution, indicating a promising direction for future research.
3. Theoretical Foundation: The theoretical analysis providing insights into the convergence behavior of EMA models adds depth and credibility to the proposed method.
4. Practical Implications: By achieving speedups of up to an order of magnitude or more, LCSC offers tangible benefits in reducing computational resources and training time, which is highly relevant in the context of large-scale model training.

**Weaknesses:**

1. Search Cost Disparity Between DM and CM: Tables 8 and 9 indicate that the search cost is more substantial for Diffusion Models (DM) compared to Consistency Models (CM), suggesting that LCSC is less effective in reducing training costs for DM.
2. The coefficients derived by the evolutionary algorithm (EA) lack interpretability, with some being significantly large (\>6) and others near zero (Fig. 8). Additionally, coefficients learned from different seeds show considerable disagreement, despite yielding similar performance improvements (as seen in Fig. 9). The lack of interpretability may hinder the understanding of the model's behavior and its generalizability. It also raises concerns about the possibility of overfitting.

**Questions:**

1. Potential Overfitting of Weights (Fig. 8): The weights depicted in Fig. 8 exhibit large absolute values (greater than 6), which might indicate overfitting. Could the authors address the following points?
- Regularization: Were any regularization techniques employed to constrain the magnitude of the coefficients?
- Interpretability: What is the theoretical or intuitive justification for assigning such large weights to certain checkpoints?
2. Convergence Curves Post-LCSC Application: Visualizing the convergence behavior after applying LCSC can provide a clearer understanding of its impact on training dynamics. Including convergence curves would help illustrate how LCSC influences the training process compared to baseline methods like EMA.
3. Convergence Curve of LCSC itself. How many iteration of evolution does it undertake before achieving a satisfactory solution?
4. Add experiments for further assessing the robustness of LCSC-Derived Coefficients.
5. Enhancing Interpretability of Coefficients: Understanding the role and significance of each coefficient can aid in demystifying the model's behavior. Implementing constraints or regularization techniques to limit the magnitude of coefficients may improve their interpretability. Alternatively, providing a theoretical justification or empirical analysis explaining the distribution of coefficient values can enhance comprehension.

---

> ### Author Response · Authors · 2024-11-25
> **Response to Reviewer Enc2 1**
>
> We sincerely thank the reviewer for their thoughtful evaluation and for highlighting the impressive experimental results, innovative findings, and theoretical foundation of our work. We greatly appreciate the recognition of LCSC’s potential to reduce computational costs and training time while improving generation quality, as well as its practical implications for large-scale model training.
>
> We have revised our paper with all updated contents in **orange**. Here are our responses to your concerns and questions:
>
> **W1:** ...LCSC is less effective in reducing training costs for DM.
> -  We acknowledge that LCSC is less effective in reducing training costs for diffusion models (DMs).
> This is because DM requires multiple forward passes to generate samples, whereas CM only requires one forward pass. This fact makes the search cost of DM much larger than that of CM.
>
> -  However, we emphasize that this limitation does not diminish the overall benefits of LCSC for DMs. As demonstrated in our paper (see Figure 1, Table 4) and supported by newly added experiments on Stable Diffusion in Section E.8, LCSC significantly enhances the final performance of late-stage or fine-tuned checkpoints derived from a converged model. Specifically, LCSC improves the quality of images generated by the DM model for the same NFE during inference. Moreover, it enables the model to achieve the same FID as the original model while requiring fewer NFEs, thereby increasing inference efficiency.
>
> -  This improvement is practically significant, as computational costs during deployment are often much higher than those during training. We have further clarified these practical advantages of LCSC for DMs in the revised paper.
>
> **W2:** The coefficients derived by the evolutionary algorithm (EA) lack interpretability, with some being significantly large ($>6$) and others near zero (Fig. 9). Additionally, coefficients learned from different seeds show considerable disagreement, despite yielding similar performance improvements (as seen in Fig. 11).. It also raises concerns about the possibility of overfitting.
>
> -  We appreciate that Reviewer ENC2 has thoughtfully broken down the main part of this concern into separate questions and presented them in the question section. Accordingly, we provide detailed responses to each question individually below.
>
> - **Q1:** Potential overfitting of weights...
>   -    We appreciate the reviewer's attention to this important concern. We would like to emphasize that our paper provides a comprehensive study demonstrating that LCSC does not suffer from overfitting, as evidenced through various analyses. Since LCSC performs the coefficient search using FID based on training data, we assessed the risk of overfitting by: (1) employing multiple evaluation metrics and (2) using test data for evaluation.
>
>   -  In particular, we conducted extensive experiments showing that models optimized with LCSC achieve consistently better results across a wide range of metrics, as discussed in Section  5.6. Specifically, we utilized metrics that differ from FID in several respects, including feature extractors (FCD, PickScore, ImageReward), calculation formulas (IS, Precision, Recall, KID, PickScore, ImageReward), and application scenarios (PickScore and ImageReward for human-based evaluation). Most metrics demonstrate consistent improvements, highlighting the generalizability of LCSC across diverse evaluation criteria.
>
>   -  Furthermore, we evaluated the model's performance using a variety of initial noise groups for generation and calculated FID based on test datasets, as detailed in Section  5.6. These evaluations consistently show improvements across metrics. Additionally, we provided numerous examples of generated images to showcase the enhanced quality of the generations (see Figures 2 and 16–22 in our paper).
>
>   -  These results collectively demonstrate that LCSC does not exhibit overfitting and is effective in identifying a generally superior model.

---

> > ### Comment · Reviewer_Enc2 · 2024-11-27
> > **Questions Addressed**
> >
> > I sincerely appreciate the authors' thoughtful efforts in addressing my questions and concerns. I am glad to see that most of my issues have been satisfactorily resolved. As a result, I will maintain my original score and continue to recommend your paper for acceptance. Thank you for your detailed responses and clarifications!

---

> ### Author Response · Authors · 2024-11-25
> **Response to Reviewer Enc2 2**
>
> -  **Q1 & Q5:** Implementing constraints or regularization techniques to limit the magnitude of coefficients
>     -    Thank you for the constructive suggestion. We conducted additional experiments to evaluate the impact of regularization. Specifically, during the search process, we clipped all coefficients to be below 1 (i.e., constraining the search results to a restricted space). Starting from the same initialization and using a consistent random seed, Figure 14 in the paper shows that LCSC produces a set of smaller and more homogeneous coefficients with regularization compared to those obtained without regularization. However, as indicated in the figure caption, applying regularization results in worse performance, suggesting that forcing LCSC to search for smaller and more homogeneous coefficients is not beneficial.
>
>     -    Notably, we observe that better performance may still be achievable with smaller and more homogeneous coefficients, although evolutionary search (ES) may not be well-suited for regularization in this application scenario. To demonstrate that, we average the searched coefficients across multiple random seeds. As shown in Figure 12, the averaged coefficients become smaller and more homogeneous, while the resulting model performs similarly well as every single searched model as indicated in the figure caption. We provide justification of coefficients with large magnitude and ES behavior in the following responses.
>
> -  **Q1 & Q5:** What is the theoretical or intuitive justification for assigning such large weights to certain checkpoints & ...providing a theoretical justification explaining the distribution of coefficients values can enhance comprehension.
>     -  First, we compared the performance of models with large positive coefficients to those with large negative coefficients. Evaluation results in Figure 10 indicate that models with large positive coefficients often achieve better FID scores, and models with small negative coefficients often achieve worse FID scores. These suggest that LCSC tends to identify the performance of each checkpoint and assign coefficients accordingly.
>     -  We also wonder whether such a coefficient pattern is necessary to get good results. In our previous response, we provided justification for the presence of large coefficients and demonstrated through experiments with averaged coefficients across different random seeds that smaller, more homogeneous coefficients can also achieve good performance. This observation suggests that having large coefficients is *not* a necessary condition for achieving good results.
>     -  Next, we delve further into why LCSC prefers such a solution that has large coefficients on a subset of checkpoints while having near-zero coefficients on most other checkpoints. We hypothesize that this phenomenon might be due to our implementation choice of the evolutionary search algorithm. Specifically, during the mutation phase, the standard deviation of the Gaussian noise we added to each *coefficient* is proportional to its current scale. Therefore, large coefficients tend to move faster, and small coefficients tend to stay stably small. This would make "smaller, more homogeneous coefficients" less likely to happen than coefficients with sparse patterns.

---

> ### Author Response · Authors · 2024-11-25
> **Response to Reviewer Enc2 3**
>
> -  **Q4:** Add experiments for further assessing the robustness of LCSC derived coefficients
>     -  From what we understand, the "robustness" in your question refers to the stability, validity and correctness of the search pattern with large coefficients. In other words, the question is about whether such a pattern is reasonable or it is just overfitting to some irrelative signals. We discuss this question in the below paragraphs. Or if the "robustness" refers to the consistency of different random seeds, the discussion can be found in the next response, where we discuss why the search patterns with respect to different random seeds could be different. If we misunderstand the meaning you intended to convey with "robustness", please let us know. We are open to further discussion.
>
>     -  To address this concern regarding the large magnitude of coefficients shown in Figure 9, we conducted additional experiments to demonstrate that these large coefficients represent meaningful optimization signals rather than accumulated noise. Specifically, we progressively scale down coefficients larger than 1.5 identified by LCSC. To ensure the resulting model remained valid, we rescaled the remaining coefficients to maintain their sum at one. The results of these experiments confirm that the large coefficients are driven by valid optimization signals rather than noise. As shown in the accompanying table, the FID of the resulting model worsens as the large coefficients are scaled down, indicating that these coefficients contribute significantly to improved model performance.
>     | | | | | | | | | | | | |
>     |-|-|-|-|-|-|-|-|-|-|-|-|
>     | Scale Down Ratio | 1.0   | 0.9   | 0.8   | 0.7   | 0.6   | 0.5   | 0.4   | 0.3   | 0.2   | 0.1   | 0.05  |
>     | | | | | | | | | | | | |
>     | FID              | 10.47 | 10.61 | 10.81 | 11.10 | 11.48 | 11.92 | 12.47 | 13.11 | 13.86 | 14.72 | 15.20 |
>     | | | | | | | | | | | | |
>
>     -  Furthermore, we have clarified our comprehensive study of overfitting risk in the previous response. Together, these studies and the accompanying empirical evidence support the validity and correctness of the coefficients derived by LCSC.
>
> -  **W2:** ...coefficients learned from different seeds show considerable disagreement...
>     -  This concern is unique to Weakness 2 and does not apply to Questions, so we address it here. We would like to highlight that previous studies have shown the existence of multiple local optima with similar test errors when optimizing high-dimensional vectors [1][2][3], such as in neural network training. We hypothesize that a similar phenomenon may be occurring in our case: LCSC might converge to different local optima for different random seeds, resulting in distinct coefficient patterns despite achieving comparable FID scores.
>
>     [1] Choromanska et al., The Loss Surfaces of Multilayer Networks
>
>     [2] Kawaguchi, Deep Learning without Poor Local Minima
>
>     [3] Garipov et al., Loss Surfaces, Mode Connectivity, and Fast Ensembling of DNNs
>
> **Q2:** ...Including convergence curves would help illustrate how LCSC influences the training process compared to baseline methods like EMA.
> -  Thank you reviewer for the constructive suggestion. We have already included several training curves in Figure 1 in the initial submission. For more comprehensive demonstration, we further include a plot showing the convergence curve of LCSC's searched model in Figure 15(a). At each point, corresponding to a specific number of training iterations, we perform LCSC with 2K search iterations using the recent 200 checkpoints available at that point. The results indicate that as the number of training iterations increases, the models searched by LCSC converge to progressively lower FID values. Notably, the convergence curve of LCSC exhibits a similar trend to that of the EMA model's convergence curve. However, LCSC consistently achieves lower FID values compared to the EMA model, highlighting its enhanced ability to effectively merge historical checkpoints.
>
> **Q3:** Convergence Curve of LCSC itself. How many iteration of evolution does it undertake before achieving a satisfactory solution?
> -  Thank you for the constructive suggestion. We have added Figure 15(b) showing the convergence curve of LCSC itself. This curve consistently reduces as the number of search iteration increases. In this paper, we set the number of search iterations to 2K, which already yields significant performance improvements, as demonstrated by our experimental results.

---

### Official Review · Reviewer_odMx · 2024-11-04

**Soundness:** 3
**Presentation:** 2
**Contribution:** 3
**Rating:** 6
**Confidence:** 2

**Summary:**

The paper proposes a method named LCSC for training diffusion models by linearly combining the intermediate weight checkpoints using an evolutionary algorithm. LSCS can significantly reduce the training cost of a diffusion model without sacrificing the generation quality. In addition, a pre-trained diffusion model can achieve better performance by fine-tuning with LCSC with few training iterations. The numerous experiment results are provided to demonstrate the effectiveness of LCSC in different datasets.

**Strengths:**

(1) The motivation is clear and easy to understand. The authors also provide theoretical analysis about why LCSC is better than EMA.

(2) The method is simple and straightforward, which uses the evolutionary search to linearly combine checkpoint weights. It seems that LCSC can be easy to implement and applied with different diffusion models.

(3) As shown in the experiment results, LCSC can significantly reduce the training cost of a diffusion model and enhance the pre-trained diffusion models.

(4) The authors provide a lot of experiment results to study LCSC from different aspects.

**Weaknesses:**

(1) It seems that the experiments about reducing the training cost only focus on datasets with small resolution (CIFAR-10 and ImageNet-64). I understand that the cost for training a diffusion model on a dataset with large resolution (e.g, LSUN-bedroom, COCO) from scratch is more expensive. I believe that such a set of experiments can make the LCSC more attractive.

(2) In my opinion, the experiment results about text-to-image generation are not convincible enough to validate the effectiveness of LCSC. The number of training iterations for fine-tuning LCM with LoRA is very small, which is only 6K with the batch size of 12. The authors should use a larger number of training iterations, or show that the generation quality of LCM will not improve with further training. I'm wondering whether the experimental settings for evaluating LCM with LCSC is suitable, since the generated images (Figure 2 and Figure 10) are in anime style while CC12M and MS-COCO are not. Besides the LCM, the authors could also fine-tune the pre-trained Stable Diffusion v1.5 for a few iterations with LCSC to see if LCSC can further enhance it.

In conclusion, I think LCSC is a good work for its motivation and method. While the experiments could be further improved and organized.

**Questions:**

(1) Can LCSC reduce the training cost of DM (not CM)? I only find experiments about reducing training cost with consistency distillation and consistency training but not vanilla DDPM.

(2) It seems that most experiments are based on CM. Do you think LCSC is more suitable for CM and CD compared with DM? And why?

(3) Could you provide the CLIP-score for text-to-image generation?

---

> ### Author Response · Authors · 2024-11-25
> **Response to Reviewer odMx 1**
>
> We sincerely thank the reviewer for their thoughtful comments and for recognizing the clarity of our motivation, the theoretical analysis comparing LCSC with EMA, and the simplicity and applicability of our method. We appreciate the acknowledgment of LCSC’s effectiveness in reducing training costs, enhancing pre-trained models, and the extensive experimental results we provided to validate our approach.
>
> We have revised our paper with all updated contents in **orange**. Here are our responses to your concerns and questions:
>
> **W1:**...experiments about reducing the training cost only focus on datasets with small resolution...
> -  We thank the reviewer for their understanding. We agree that training diffusion models on high-resolution datasets would enhance the appeal of our work. However, we lack access to sufficient computational resources to support such experiments.
> -  Nonetheless, we would like to point out several points to address your concerns: **(1)** Generally speaking, training high-resolution diffusion models with large parameter sizes is impractical for most users. In most cases, such models are trained from scratch by large organizations, and many users rely on publicly released versions. This scenario aligns closely with the second use case of LCSC, where users need only fine-tuning the released model for a few iterations before applying LCSC to enhance it. The effectiveness of LCSC with large models in such cases is demonstrated in Tables 3, 5, and 6. **(2)**  While the resolution of ImageNet-64 is not high, it remains a diverse dataset with 1,000 classes and over one million images. Label-conditional generation on ImageNet-64 is a challenging task, as the original CT model achieves an FID of only around 13 with 1-step generation. As shown in Tables 2 and 5, LCSC achieves remarkable training speedup and performance improvements on this dataset. Therefore, it is reasonable to expect that LCSC would deliver significant training acceleration and performance gains on more complex, high-resolution tasks.
>
> **W2:** ...the experiment results about text-to-image generation are not convincible enough...
>
> -  Thank you for pointing out these questions. Below are our response to each specific question.
> -  **W2.1:** ...The number of training iterations for fine-tuning LCM with LoRA is very small...
>     -   Our setting follows the official default setting of training LCM-LoRA in https://github.com/luosiallen/latent-consistency-model/tree/main/LCM_Training_Script/consistency_distillation#example-with-laion-a6-dataset-1, which suggests a training configuration with **1k steps** and a **batch size of 12**. To obtain more high-quality LoRA checkpoints, we further train the model for 6k steps.
>     -   As discussed in Section  4, LCSC can be applied at **arbitrary positions** along the training trajectory to enhance the model. Our experiments have already shown that LCSC can be applied to the checkpoints near 6k iterations and achieve better performance. Therefore, we think our existing experiments are valid to prove the effectiveness of LCSC.
>    - To further address the concerns and make the experiments more comprehensive, we follow your suggestions to train LCM-LoRA for more iterations and choose a FID-converged point to apply LCSC. The training curve is shown as below.
> | | | | | |
> |-|-|-|-|-|
> |-|6k|20k|40k|50k|
> | | | | | |
> |FID|32.52|25.15|24.74|24.69|
> | | | | | |
>
>         We then choose the checkpoints between 34k and 40k steps with an interval 20 to conduct LCSC. The results are demonstrated below. LCSC achieves better results across various metrics compared to the vanilla model, while the training trajectory has already converged.
>
>         | | | | | |
>         |-|-|-|-|-|
>         | |FID|PickScore|Wining Rate@PKS|CLIP Score|
>         | | | | | | |
>         |LCSC|**23.53**|**0.52**|**60%**|**26.43**|
>         |Vanilla Training|24.74|0.48|40%|26.22|
>         | | | | | |

---

> ### Author Response · Authors · 2024-11-25
> **Response to Reviewer odMx 2**
>
> - **W2.2:** ...whether the experimental settings for evaluating LCM with LCSC is suitable...
>   - **Why we use such settings:** For model inference, we follow the official inference setting of LCM-LoRA in https://huggingface.co/latent-consistency/lcm-lora-sdv1-5#text-to-image, using the dreamshaper-7 model as the backbone for better visualization quality. This is the reason for generating anime-style images. For evaluation, since MS-COCO is the most popular benchmark for evaluating text-to-image models and CC12M is our training set, we conduct search on CC12M and evaluate on MS-COCO.
>   - **Whether this setting is suitable:** Although there may be some misalignment between the styles of generated images and the evaluation dataset, our evaluation metrics could also capture other aspects of image quality such as clarity, content coherence, and so on. Therefore, we think our evaluation setting can provide useful insights into the image quality of LCSC. Additionally, we would like to point out that LCSC is compatible with any evaluation metrics. If the reviewer has other suggestions on the evaluation settings (including the metrics and datasets for search and evaluation), we are happy to add the experiments.
>   - **LCSC has indeed optimized the image quality:** From the visualizations in Figure 2 and Figure 16, we can find that the generated images by LCSC are more semantically coherent compared to the baseline (e.g., the front wheels in Fig. 2 (left) and the sitting posture in Fig. 2 (right)), and have richer details. Moreover, the metrics we use (i.e., PickScore and ImageReward) are both universally human aesthetic oriented. Therefore, the quantitative results, which show that LCSC outperforms the baseline over these metrics, indicate the actual improvement in image quality.
>
> -  **W2.3:** ...the authors could also fine-tune the pre-trained Stable Diffusion v1.5 for a few iterations with LCSC to see if LCSC can further enhance it.}
>     -  This is a great suggestion for our work! We follow your advice and fine-tune the Stable-Diffusion-v1.5 checkpoints on CC12M with LoRA for 20k iterations. Then we applied LCSC to checkpoints with an interval of 100. We use 15-step DPM-Solver for searching, and evaluate the performance with both 10-step and 15-step DPM-Solver on MS-COCO. The results are shown in the table below. We additionally test the PickScore between the SDv1-5 model with 15 NFE and LCSC with 10 NFE, getting 0.49 and 0.51, respectively, with a winning rate of 57\% for LCSC. The conclusions in Section  5.3 still holds, where LCSC enhances the performance of Stable-Diffusion-v1.5, while achieving 1.5$\times$ inference acceleration ratio. We have included these results in our revised paper.
> | | | | | | |
> |-|-|-|-|-|-|
> | |NFE|FID|PickScore|Winning rate@PKS|CLIP Score|
> | | | | | | |
> |LCSC|15|**16.30**|**0.53**(v.s. SD)/**0.53**(v.s. tuning)|**56**(v.s. SD)/**55**(v.s. tuning)|**26.69**|
> |SDv1-5|15|17.55|0.47|44|26.60|
> |Vanilla tuning|15|17.05|0.47|45|26.61|
> |LCSC|10|**16.68**|**0.59**(v.s. SD)/**0.51**(v.s. tuning)|**64**(v.s. SD)/**53**(v.s. tuning)|**26.61**|
> |SDv1-5|10|18.16|0.41|36|26.57|
> |Vanilla tuning|10|17.35|0.49|47|26.56|
> | | | | | | |
>
> **Q1:** Can LCSC reduce the training cost of DM (not CM)?} We acknowledge that LCSC is less effective in reducing the training cost of DM.
>   -  Because the inference time of DM is much larger than CM, the cost of the evolutionary search on DM is much larger than that of CM, which reduces the effectiveness of LCSC in reducing the training cost.
>
>   -  However, we emphasize that this limitation does not diminish the overall benefits of LCSC for DMs. As demonstrated in our paper (see Figure 1 and Table 4) and supported by newly added experiments on Stable Diffusion, LCSC significantly enhances the final performance of late-stage or fine-tuned checkpoints derived from a converged model. Specifically, LCSC improves the quality of images generated by the DM model for the same NFE during inference. Moreover, it enables the model to achieve the same FID as the original model while requiring fewer NFEs, thereby increasing inference efficiency.
>
>   -  This improvement is practically significant, as computational costs during deployment for users are often much higher than those during training. We have further clarified these practical advantages of LCSC for DMs in the revised paper.

---

> ### Author Response · Authors · 2024-11-25
> **Response to Reviewer odMx 3**
>
> **Q2:** ...Do you think LCSC is more suitable for CM and CD compared with DM? And why?
>
> -  Due to the small search cost as discussed above, we agree that LCSC is more convenient to use with CM than with DM. Since our computational resources are limited, it is a natural choice for us to apply LCSC on CM to explore more broadly, such as validating the effectiveness of LCSC on text-to-image tasks and investigating whether LCSC can be applied with LoRA checkpoints.
>
> -  However, we believe the existing experimental results on DM (Tab. 4 and the newly added experiments on Stable-Diffusion-v1.5) are sufficient to demonstrate the effectiveness of LCSC on DM, enabling significant performance enhancement and inference acceleration.
>
> **Q3:** Could you provide the CLIP-score for text-to-image generation?
> -  We add the CLIP-score of all methods in Table 6 in the paper (as copied below). All newly added experiments have also included the CLIP-score as a metric. LCSC demonstrates superior performance in most cases. It is important to note that the CLIP-score measures semantic consistency, which can remain stable even as image quality significantly degrades. Consequently, the CLIP-score is less sensitive compared to other metrics that account for image quality. These results have been included in the revised version of our paper.
> | | | | | | |
> |-|-|-|-|-|-|
> | |Search NFE|Eval NFE|CLIP Score| | |
> | | | | | | |
> |LCSC|4|4|26.39| | |
> |Vanilla Training|-|4|26.02| | |
> | | | | | | |
> |LCSC|2|2|26.01| | |
> |LCSC|4|2|25.89| | |
> |Vanilla Training|-|2|25.16| | |
> | | | | | | |

---

> ### Author Response · Authors · 2024-11-29
> **Follow-up Comment**
>
> Thank you for your thoughtful review and your initial positive evaluation of our work. We deeply appreciate your recognition of the motivation, simplicity, and effectiveness of LCSC, as well as the value of our theoretical analysis and comprehensive experiments.
>
> Since your feedback has been instrumental in refining our presentation, we have worked diligently in our rebuttal to address the concerns and questions raised. We hope our responses and additional clarifications have further strengthened your confidence in the contributions of our work.
>
> If you find that our rebuttal has satisfactorily resolved your concerns, we would kindly ask you to consider revisiting your score to better reflect your updated assessment of the paper. We respect your discretion and remain grateful for your valuable feedback and initial positive evaluation.

---

### Official Review · Reviewer_K43d · 2024-11-07

**Soundness:** 4
**Presentation:** 3
**Contribution:** 3
**Rating:** 6
**Confidence:** 2

**Summary:**

This paper proposes LCSC (Linear Combination of Saved Checkpoints), a method to improve the performance and efficiency of diffusion and consistency models. The underlying idea is to use an optimal linear combination of model checkpoints saved during training. These optimal coefficients are determined using an evolutionary search method. The authors demonstrate that this approach can not only decrease the training cost but also enhance the performance of pre-trained models. The experiments which were performed on CIFAR-10 and ImageNet-64 provably demonstrate that this approach results in an increase in training time and improvements in evaluation metrics.

**Strengths:**

- The simple methodology presented seems to have a well-motivated theoretical basis, and is quite effective in improving the training of diffusion and consistency models. Further, this method does not require backprop and side steps the need for differentiable loss functions through the evolutionary search.
- The analysis presented on the method along with the visualizations of the landscape to demonstrate that optimal model weights lie in basins which are inaccessible by optimization but may be accessible through checkpoint averaging.
- It has clear value in reducing compute time and improving model performance as demonstrated through the experiments. Seems like there's speedups by an order of magnitude which is quite impressive.

**Weaknesses:**

See questions

**Questions:**

- Currently, the experiments seems to be limited to image generation tasks. Would it be possible to demonstrate this to another diffusion task such as audio or video?
- Have you studied how these coefficients change over the course of training? There perhaps maybe an interpretability study to understand the importance of different stages of training on the best performance
- On a similar front, can it be extended to layers? That is, composition of weights on layers may differ from the composition of overall weights and perhaps lead to a better result?

---

> ### Author Response · Authors · 2024-11-25
> **Response to Reviewer K43d**
>
> We sincerely thank the reviewer for their thoughtful feedback and for highlighting the theoretical soundness, simplicity, and practical value of LCSC. We appreciate the recognition of our method’s effectiveness in reducing compute time, improving model performance, and accessing optimal solutions through checkpoint averaging.
>
> We have revised our paper with all updated contents in **orange**. Here are our responses to your concerns and questions:
>
> **Q1:** Conduct experiments on another diffusion tasks such as audio or video.
> -  We thank the reviewer for their thoughtful question and agree that extending the experiments to different scenarios is indeed an interesting direction. We believe that LCSC would be effective for diffusion tasks across other modalities, as our method relies solely on checkpoints from the training phase, and diffusion tasks generally exhibit similar training dynamics. In this work, we focus on image generation as a proof of concept, prioritizing theoretical analysis and comprehensive experiments on various types of diffusion models. We consider applications to other modalities as a promising avenue for future work.
>
> **Q2:** How coefficients change over the course of training? Is there an interpretability study to understand the importance of different stages of training on the best performance?
> -  Since we include only checkpoints from the later stages of training in the list, the checkpoints included in the search list for LCSC vary throughout the training process. This makes it challenging to analyze the patterns in the coefficients for specific checkpoints.
> Furthermore, as demonstrated in Figure 11 of our paper, even with a fixed set of checkpoints, the coefficients vary depending on the search seed. These factors make it difficult to track how the coefficient of a specific checkpoint evolves during training.
>
> -  However, we consistently observe that the coefficients exhibit a sparse pattern. Additionally, because LCSC consistently focuses on the most recent checkpoints, we do not observe significant differences in the coefficients across different training stages.
>
> **Q3:** Can the method extend to the composition of weights on layers?
> -  Using different coefficients across layers to compose weights is indeed an interesting and promising direction, as it expands the diversity of the search space. To validate this idea, we conducted an experiment on CIFAR-10 CD at 250k iterations. We grouped all weights according to their corresponding resolution in the UNet, assigning different combination coefficients to each group. This approach resulted in performance improvements in FID, as shown below.
> | | | |
> |-|-|-|
> | |Grouped Combination|Overall Combination|
> |FID| **2.43** | 2.77 |
> | | | |
> - This result indicates that using different coefficients across layers is indeed a promising direction.
> However, extending weight composition to layers significantly increases the size of the search space, making evolutionary search more challenging. Developing more efficient methods to adapt our approach for layer-wise composition represents an exciting direction for future research.

---

> > ### Comment · Reviewer_K43d · 2024-11-25
> > **Questions Addressed**
> >
> > Thank you for your detailed rebuttal. You have addressed my questions and concerns. I will maintain my original score and recommendation for acceptance.

---

### Meta-Review · Area_Chair_G8cS · 2024-12-19

**Metareview:**

This paper proposes to leverage an optimal linear combination of intermediate model checkpoints during training to improve the efficiency (training cost) of diffusion and consistency models while maintaining or improving their performance.

Four knowledgeable referees reviewed this submission and acknowledged that the approach was well motivated from a theoretical perspective (K43d, Enc2,AvHi), and presented in a clear and easy to follow way (odMx). The reviewers also found the method novel (AvHi), interesting (AvHi), and simple (K43d, odMx), and valuable to reduce the training costs while improving model performance (K43d, odMx)

The main concerns raised by the reviewers were:
1. Experiments were not found fully convincing: limited experiments to image generation tasks (K43d), low resolution datasets (odMx), unclear whether generation quality would improve with additional training (odMx), some claims requiring additional evidence (AvHi)
2. Missing analysis of how coefficients change over the course of training (K43d), and of their interpretability (Enc2)

The reviewers also asked clarifying questions related to the metrics landscape (AvHi), and the extension of the proposed approach to layers (K43d). During rebuttal and discussion, the authors discussed image generation as a proof of concept for their approach, described the challenges in analyzing the evolution of coefficients and gave some intuitions, showcased the potential of the method to be applied layer-wise through one experiments, presented results with increased numbers of iterations, added model finetuning experiments, added comparisons based on CLIPscore, and added experiments to address concerns w.r.t. the magnitude of the coefficients. The discussion period addressed most of the reviewers' concerns, who unanimously lean towards acceptance. The AC agrees with the reviewers' assessment and therefore recommends to accept.

**Additional Comments On Reviewer Discussion:**

All details are captured in the meta-review.

---

### Decision · Program_Chairs · 2025-01-22

Accept (Poster)